# A Geometric Approach to Personalized Recommendation with Set-Theoretic Constraints Using Box Embeddings

**Shib Dasgupta** [1]   **Michael Boratko** [1]   **Andrew McCallum** [1]

## Abstract

Personalized item recommendation typically suffers from data sparsity, which is most often addressed by learning vector representations of users and items via low-rank matrix factorization. While this effectively densifies the matrix by assuming users and movies can be represented by linearly dependent latent features, it does not capture more complicated interactions. For example, vector representations struggle with set-theoretic relationships, such as negation and intersection, *e.g.* recommending a movie that is "comedy and action, but not romance". In this work, we formulate the problem of personalized item recommendation as matrix completion where rows are *set-theoretically dependent*. To capture this set-theoretic dependence we represent each user and attribute by a hyper-rectangle or *box* (*i.e.* a Cartesian product of intervals). Box embeddings can intuitively be understood as trainable Venn diagrams, and thus not only inherently represent similarity (via the Jaccard index), but also naturally and faithfully support arbitrary set-theoretic relationships. Queries involving set-theoretic constraints can be efficiently computed directly on the embedding space by performing geometric operations on the representations. We empirically demonstrate the superiority of box embeddings over vector-based neural methods on both simple and complex item recommendation queries by up to 30% overall.

## 1. Introduction

Recommendation systems are a standard component of most online platforms, providing personalized suggestions for products, movies, articles, and more. In addition to generic recommendation, these platforms often present the option for the user to search for items, either via natural language or structured queries. While collaborative filtering methods like matrix factorization have proven successful in addressing data sparsity for unconditional generation, they often fall short when attempting to combine them with more complicated queries. This is not unexpected, as vector embeddings, while effectively capturing linear relationships, are ill-equipped to handle the complex set-theoretic relationships. Even advanced neural network-based approaches, which are designed to capture intricate relationships, have been shown to struggle with set-theoretic compositionality that underlie many real-world preferences.

Let us consider an example where a user named Bob wants to watch a comedy which is not a romantic comedy. Assuming we have a prior watch history for users, standard collaborative filtering techniques (e.g. low-rank matrix factorization) would yield a learned score function $\text{score}(m, \texttt{Bob})$ for each movie $m$. If we also have movie-attribute annotations, we could form the set of comedies $C$ and set of romance movies $R$ and simply filter to those movies in $C \setminus R$, however this assumes that the movie-attribute annotations are complete, which is rarely the case in practice. In fact, Dasgupta et al. (2023) show that item-attribute matrices, even manually curated, are often incomplete, and remain sparse and noisy due to limited coverage and annotation effort.

A standard approach in a setting with sparse data is to learn a low-rank approximation for the attribute $\times$ movie matrix $\mathbf{A}$, yielding a dense matrix $\hat{\mathbf{A}}$. We can then form sets of movies based on this dense matrix using an (attribute-specific) threshold, *e.g.* $\hat{C} := \{m \mid \hat{A}_{\text{comedy},m} > \tau_{\text{comedy}}\}$ and $\hat{R} := \{m \mid \hat{A}_{\text{romance},m} > \tau_{\text{romance}}\}$, and then rank movies $m \in \hat{C} \setminus \hat{R}$ according to $\text{score}(m, \texttt{Bob})$. While this approach does allow for performing the sort of queries we are after, it suffers from three fundamental issues:

1. Limited user-attribute interaction: Since the attribute classification is done independently from the user, any latent relationships between the user and attribute cannot be taken into account.

---

[1]Manning College of Information & Computer Sciences, UMass Amherst. Correspondence to: Shib Dasgupta <ssdasgupta@cs.umass.edu>.

*Proceedings of the $42^{nd}$ International Conference on Machine Learning*, Vancouver, Canada. PMLR 267, 2025. Copyright 2025 by the author(s).

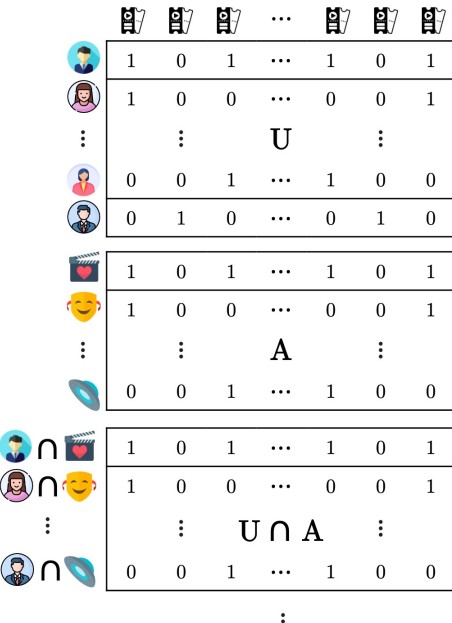

Figure 1: Standard matrix completion assumes you are given partial information about the user × movie matrix **U**, and potentially incomplete information about the attribute × movie matrix **A**.

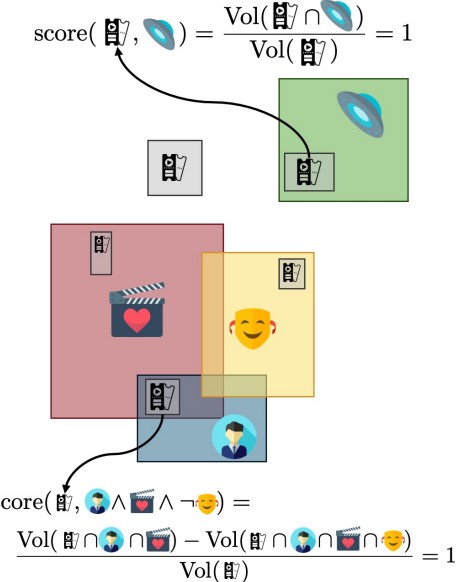

Figure 2: Box embeddings represent the movies, users, and attributes as "boxes" (Cartesian products of intervals) in $\mathbb{R}^n$.

2. Error compounding: Errors in the completion of attribute sets accumulate as the number of sets involved in the query increase.

3. Mismatched inductive-bias: Our queries can be viewed as set-theoretic combinations of the rows, not linear combinations. As such, using a low-rank approximation of the matrix may be misaligned with the eventual use.

In this paper, we formulate the problem of attribute-specific recommendation as matrix completion where rows are not necessarily *linear combinations* of each other but, rather, are *set-theoretic combinations* of each other. More precisely, given some user × movie interaction matrix **U** and attribute × movie matrix **A**, the queries we are considering are set-theoretic combinations of these rows (see Figure 1). For example, the ground-truth data for comedies which are not romance movies which Bob likes would be the vector $x \in \{0,1\}^{|M|}$, where $x_m = 1$ if and only if $\mathbf{U}_{\text{Bob},m} = 1$ and $\mathbf{A}_{\text{comedy},m} = 1$ and $\mathbf{A}_{\text{romance},m} = 0$. Note that this is not a linear combination of the previous rows, and so while the inductive bias of low-rank factorization has proven immensely effective for collaborative filtering we should not expect it to be directly applicable in this setting.

Instead, we propose to learn representations for the users and attributes that are consistent with specific set-theoretic axioms. These representations must also be compactly parameterizable in a lower-dimensional space, differentiable with respect to some appropriate score function, and allow for efficient computation of various set operations. Box Embeddings (Vilnis et al., 2018; Dasgupta et al., 2020), which are axis-parallel $n$-dimensional hyperrectangles, meet these criteria (see Figure 2). The volume of a box is easily calculated as the product of its side-lengths. Furthermore, box embeddings are closed under intersection (*i.e.* the intersection of two boxes is another box). Inclusion-exclusion thus allows us to calculate the volume of arbitrary set-theoretic combinations of boxes.

The contributions of our paper are as follows -

1. We model the problem of attribute-specific query recommendation as "set-theoretic matrix completion", where attributes and users are treated as sets of items. We discuss the challenges faced by existing machine-learning approaches for this problem setup.

2. We demonstrate the inconsistency of existing vector embedding models for this task. Additionally, we establish box embeddings as a suitable embedding method for addressing such set-theoretic problems.

3. We conduct an extensive empirical study comparing various vector and box embedding models for the task of set-theoretic query recommendation.

Box embeddings, with their geometric set operations, significantly outperform all vector-based methods. We also evaluate score multiplication and threshold-based prediction for both vector and box embedding models, and find that performing set operations directly on the box embeddings performs best, solidifying our claim that the inductive bias of box embeddings provides the necessary generalization capabilities to address set-theoretic queries.

## 2. Task Formulation

### 2.1. Background

Matrix completion is a fundamental problem in machine learning, and arises in a wide array of tasks, from recommender systems to image reconstruction. Formally, this problem is typically modeled as follows: Given a matrix $X \in \mathbb{R}^{m \times n}$ where only a subset of the entries are observed, find a complete matrix $\hat{X} \in \mathbb{R}^{m \times n}$ which closely approximates $X$ on the observed entries. For the task of recommendation, this involves predicting user interactions with items they have not previously interacted with, and a common assumption is that the preferences of users and characteristics of the items can be expressed by a small number of latent factors, with the alignment of these latent factors captured via dot-product. This justifies the search for a low-rank approximation $\hat{X} = BC$, where $B \in \mathbb{R}^{m \times D}$ and $C \in \mathbb{R}^{D \times n}$. In the case where the original matrix is binary, $X \in \{0, 1\}^{m \times n}$, it is common to perform *logistic matrix factorization*, where an elementwise sigmoid is applied after the dot-product of latent factors, which we denote (with slight abuse of notation) as $\hat{X} = \sigma(BC)$.

### 2.2. Set-Theoretic Matrix Completion

We will describe the task of set-theoretic matrix completion on the setting of movies, users, and attributes, though the formulation and our proposed model can be generalized to arbitrary domains. We are given a set $\mathcal{D}_U \subseteq U \times M$ of user-movie interactions, and a set $\mathcal{D}_A \subseteq A \times M$ of attribute-movie pairs. We assume both of these sets are incomplete.

Our goal is to eventually be able to recommend movies based on some query, for example "comedy and not romance". Such a query for a particular user can be represented as $u \wedge a_1 \wedge \neg a_2$, where $u$ is the user, $a_1 = $ comedy and $a_2 = $ romance. We let $Q$ be the set of all queries of interest, which depends on which queries we anticipate evaluating at inference time. In this work, we will take $Q$ to be queries of the form $u$, $a_1$, $u \wedge a_1$, $u \wedge a_1 \wedge a_2$, and $u \wedge a_1 \wedge \neg a_2$, where $u \in U$ and $a_1, a_2 \in A$.

With this formulation, we can view our task as matrix completion for a matrix $X \in \{0, 1\}^{|Q| \times |M|}$, where the rows are derived by applying bitwise operators on the rows of

user and attribute data. While we could, in theory, proceed directly with logistic matrix factorization on this matrix, there are both practical and theoretical reasons to search for an alternative. First, the number of rows of this matrix is very large relative to the original data - in our case we have $|Q| = \mathcal{O}(|U||A|^2)$, but in general $|Q| = \mathcal{O}(3^{|U||A|})$. This poses practical issues, both at training time (as there are an exponential number of elements of $X$ to traverse) and inference time (storing the low-rank approximations requires $\mathcal{O}(|Q|)$ memory, which is much larger than $|U| + |A|$). There are also theoretical issues with the underlying assumption, as it is no longer reasonable to assume the rows of $\sigma^{-1}(X)$ are linear combinations of some latent factors.

## 3. Method

Our proposed solution to address these issues starts by defining the sets of movies which comprise the queries of interest. Let, $\mathcal{P}(M)$ be the power set of movies $M$. Specifically, for each user $u$ we can define the set $M_u = \{m \mid (u, m) \in \mathcal{D}_U\}$, and for each attribute $a$ we can define the set $M_a = \{m \mid (a, m) \in \mathcal{D}_A\}$. If we let $\mathcal{M} \subseteq \mathcal{P}(M)$ be the collection of all such sets, then the set of movies corresponding to a given query $q$ are direct set-theoretic combinations of elements in $\mathcal{M}$. Hence, the reasonable underlying assumption, in this case, is to model the elements of $\mathcal{M}$ as sets via a map $f : \mathcal{M} \to R$ where $R$ is also a set of sets, and the map $f$ respects set-theoretic operations, *i.e.* $f(S \cap T) = f(S) \cap f(T)$ and $f(S \setminus T) = f(S) \setminus f(T)$, etc. Such a map is referred to as a *homomorphism of Boolean algebras*, and the problem of learning such a function was explored in general in (Boratko et al., 2022). In our work, we propose box embeddings as the function $f$ which can be trained to obey the homomorphism constraints. As a result, user-attribute-item representations based on box embeddings could serve as an optimal inductive bias for the proposed set-theoretic matrix completion task.

### 3.1. Set-theoretic Representation Box Embeddings

As introduced in Vilnis et al. (2018), box embeddings represent entities by a hyperrectangle in $\mathbb{R}^D$, *i.e.* a Cartesian product of intervals. Let the box embedding for user $u$ be:

$$\text{Box}(u) = \prod_{d=1}^{D}[u_d^{\llcorner}, u_d^{\urcorner}] = [u_1^{\llcorner}, u_1^{\urcorner}] \times \ldots \times [u_D^{\llcorner}, u_D^{\urcorner}] \subseteq \mathbb{R}^D,$$

where $[u_d^{\llcorner}, u_d^{\urcorner}]$ is the interval for $d$-th dimension, $u_d^{\llcorner} < u_d^{\urcorner}$ for $d \in \{1, \ldots, D\}$.

The volume of an interval is defined as the length of the interval $\text{Vol}((u_d^{\llcorner}, u_d^{\urcorner})) = \max(u_d^{\urcorner} - u_d^{\llcorner}, 0)$.

Let, $\text{Box}(m) = \prod_{d=1}^{D}[m_d^{\llcorner}, m_d^{\urcorner}]$ be the box embeddings for a movie $m$. At dimension $d$, the volume of intersection

between user $u$ and movie $m$ is defined as -

$$\text{VolInt}((u_d^\llcorner, u_d^\urcorner), (m_d^\llcorner, m_d^\urcorner))$$
$$= \max\left(\min(u_d^\urcorner, m_d^\urcorner) - \max(u_d^\llcorner, m_d^\llcorner), 0\right).$$

When the movie interval $[m_d^\llcorner, m_d^\urcorner]$ is completely contained by user interval $[u_d^\llcorner, u_d^\urcorner]$, then $\frac{\text{VolInt}((u_d^\llcorner, u_d^\urcorner),(m_d^\llcorner, m_d^\urcorner))}{\text{Vol}((m_d^\llcorner, m_d^\urcorner))} = 1$. This objective creates a set-theoretic interpretation with box embeddings, where user $\text{Box}(u)$ contains all the movie boxes related to $u$ (Figure 2). The score for containment for a single dimension $d$ is formulated as:

$$F_{\text{Box}}((u_d^\llcorner, u_d^\urcorner), (m_d^\llcorner, m_d^\urcorner))$$
$$:= \frac{\text{VolInt}((u_d^\llcorner, u_d^\urcorner), (m_d^\llcorner, m_d^\urcorner))}{\text{Vol}((m_d^\llcorner, m_d^\urcorner))}$$
$$:= \frac{\max(\min(u_d^\urcorner, m_d^\urcorner) - \max(u_d^\llcorner, m_d^\llcorner), 0)}{\max(m_d^\urcorner - m_d^\llcorner, 0)}. \quad (1)$$

The overall containment score is the multiplication of $F_{\text{Box}}$ for each dimension. The $\log$ of this score is referred to as the energy function as given:

$$\text{E}_{\text{Box}}(u, m) := -\log \prod_{d=1}^{D} F_{\text{Box}}((u_d^\llcorner, u_d^\urcorner), (m_d^\llcorner, m_d^\urcorner)). \quad (2)$$

This energy function is minimized when the user $\text{Box}(u)$ contains the movie $\text{Box}(m)$. Previous works have highlighted the difficulty of optimizing an objective including these hard $\min$ and $\max$ functions (Li et al., 2019; Dasgupta et al., 2020). In our work, we use the latter solution, termed GUMBELBOX, which treats the endpoints $x^\llcorner$ and $x^\urcorner$ as mean of GumbelMax and GumbelMin random variables, respectively. Given 1-dimensional box parameters $\{[x_n^\llcorner, x_n^\urcorner]\}_{n=1}^{N}$, we define the associated GumbelMax random variables $X_n^\llcorner$ with mean $x_n^\llcorner$ and scale $\beta$, as well as the GumbelMin random variables $X_n^\urcorner$ with mean $x_n^\urcorner$ and scale $\beta$. Dasgupta et al. (2020) calculates that the expected volume of intersection of intervals $\{[X_n^\llcorner, X_n^\urcorner]\}$ can be approximated by

$$\mathbb{E}\left[\max\left(\min_n X_n^\urcorner - \max_n X_n^\llcorner, 0\right)\right]$$
$$\approx \text{LSE}_\beta\left(\text{LSE}_{-\beta}(x_1^\urcorner, \ldots, x_N^\urcorner) - \text{LSE}_\beta(x_1^\llcorner, \ldots, x_N^\llcorner), 0\right).$$

essentially replacing the hard $\min$ and $\max$ operators with a smooth approximation, $\text{LSE}_t(\mathbf{x}) := t\log(\sum_i e^{x_i/t})$. Expected intersection volume in higher dimensions is just a product of the preceding equation, as the random variables are independent. We use this GUMBELBOX (abbrev $GB$) formulation in our work changing the notations $F_{Box}, \text{Vol}, \text{VolInt}$ to $F_{GB}, \text{Vol}_{GB}, \text{VolInt}_{GB}$. We modify the per-dimension score function $F_{\text{Box}}$ in (2) by replacing

the ratio of hard volume calculations with the approximation to the expected volume,

$$F_{\text{GB}}((u_d^\llcorner, u_d^\urcorner), (m_d^\llcorner, m_d^\urcorner); (\tau, \nu))$$
$$:= \frac{\text{LSE}_\nu(\text{LSE}_{-\tau}(u_d^\urcorner, m_d^\urcorner) - \text{LSE}_\tau(u_d^\llcorner, m_d^\llcorner), 0)}{\text{LSE}_\nu(m_d^\urcorner - m_d^\llcorner, 0)}$$
$$=: \frac{\text{VolInt}_{\text{GB}}((u_d^\llcorner, u_d^\urcorner), (m_d^\llcorner, m_d^\urcorner); (\tau, \nu))}{\text{Vol}_{\text{GB}}((m_d^\urcorner - m_d^\llcorner); \nu)}. \quad (3)$$

### 3.2. Training

We model each user, attribute, and movie as a box in $\mathbb{R}^D$, and denote the map from these entities to their associated box parameters as $\theta$, i.e., the trainable box embedding for user $u$ is $\theta(u) := \text{Box}(u)$. Our goal is to train these box representations to represent certain sets of movies which allow us to perform the sort of queries we are interested in. As motivated above, for a given user $u$, we train $\text{Box}(u)$ to approximate the set $M_u$ via a noise-contrastive estimation objective. Namely, for each $(u, m) \in \mathcal{D}_U$, we have a loss term

$$\ell_{(u,m)}(\theta) := \text{E}_{\text{GB}}(u, m; \theta)$$
$$- \mathbb{E}_{\tilde{m} \sim M}\left[\log\left(1 - \exp(-\text{E}_{\text{GB}}(u, \tilde{m}; \theta)))\right)\right].$$

The first term is minimized when $\text{Box}(u)$ contains $\text{Box}(m)$. We approximate the second term via sampling, which encourages $\text{Box}(u)$ to be disjoint from $\text{Box}(\tilde{m})$ for a uniformly randomly sampled movie $\tilde{m}$. We define an analogous loss function $\ell_{(a,m)}(\theta)$ for attribute-movie interactions, which trains $\text{Box}(a)$ to contain the box $\text{Box}(m)$ for each $m$ such that $(u, m) \in \mathcal{D}_U$.

The overall loss function is a convex combination of these loss terms:

$$\mathcal{L}(\theta; \mathcal{D}_U, \mathcal{D}_A) := w * \sum_{(u,m) \in \mathcal{D}_U} \ell_{(u,m)}(\theta)$$
$$+ (1 - w) * \sum_{(a,m) \in \mathcal{D}_A} \ell_{(a,m)}(\theta).$$

for a hyperparameter $w \in [0, 1]$. This optimization ensures that the movie boxes are contained within the corresponding user and attribute boxes, thereby establishing a set-theoretic inductive bias. Both numbers of negative samples and $w$ are hyperparameters for training (Please Refer to Section 4, Appendix A.2) for further details. Training box embeddings is generally efficient, as the computation of box intersection volumes can be parallelized across dimensions. We provide training time details for the box embedding model and other vector-based baselines in Table 11 in Appendix C

### 3.3. Inference

During inference, given the trained embedding model $\theta$ and a user $u$ we determine the user's preference for the movie

$m$ by negating and exponentiating the energy function,

$$\text{score}(m, u; \theta) := \exp\left(-\,\text{E}_{\text{GB}}(u, m; \theta)\right)$$

$$= \prod_{d=1}^{D} F_{\text{GB}}\left(\theta(u)_d, \theta(m)_d; (\tau, \nu)\right) \in \mathbb{R}_{\geq 0},$$

where $\theta(x)_d = (x_d^{\llcorner}, x_d^{\urcorner})$. Since the calculation is simply a product over dimensions, for notational clarity we will restrict our discussion for more complex queries to the one-dimensional case, and omit the explicit dependence on temperature hyperparameters, so

$$\text{score}(m, u; \theta) := \frac{\text{VolInt}_{\text{GB}}\left(\theta(m), \theta(u)\right)}{\text{Vol}_{\text{GB}}\left(\theta(m)\right)}$$

which is the *proportion of $\theta(m)$ which is contained within $\theta(u)$* (see Figure 2). It achieves it's maximum at 1 if $\theta(u)$ contains $\theta(m)$, and is minimized at 0 when they are disjoint, corresponding to the motivation that $\theta(u)$ represents the set of movies that user $u$ has interacted with.

Given a query with a conjunction between attributes (*e.g.* "comedy and action") we denote the attributes involved $a_1$ and $a_2$. Similarly to the score for a single user query, we define the score for these attributes as the proportion of the movie box $\theta(m)$ which is contained inside of the (soft) intersection of boxes $\theta(u)$, $\theta(a_1)$, and $\theta(a_2)$, *i.e.*

$$\text{score}(m, u \wedge a_1 \wedge a_2; \theta) := \frac{\text{VolInt}_{\text{GB}}\left(\theta(m), \theta(u), \theta(a_1), \theta(a_2)\right)}{\text{Vol}_{\text{GB}}\left(\theta(m)\right)}.$$

Again, this score is maximized if $\theta(m)$ is contained inside $\theta(u), \theta(a_1)$, and $\theta(a_2)$, and minimized when it is disjoint.

In order to address queries with set differences, recall that, given two measurable sets $S$ and $T$, we can compute the volume of $S \setminus T$ as $\text{Vol}(S \setminus T) = \text{Vol}(S) - \text{Vol}(S \cap T)$. Thus, if the query involves a negated attribute (*e.g.* "comedy and not action"), we define

$$\text{score}(m, u \wedge a_1 \wedge \neg a_2; \theta) := \frac{\text{VolInt}_{\text{GB}}\left(\theta(m), \theta(u), \theta(a_1)\right)}{\text{Vol}_{\text{GB}}\left(\theta(m)\right)}$$
$$- \frac{\text{VolInt}_{\text{GB}}\left(\theta(m), \theta(u), \theta(a_1), \theta(a_2)\right)}{\text{Vol}_{\text{GB}}\left(\theta(m)\right)}$$

This score is maximized when $\theta(m)$ is contained inside $\theta(u)$ and $\theta(a_1)$ while being disjoint from $\theta(a_2)$, and decreases when these conditions are not met.

Our containment-based scoring framework naturally generalizes to more complex logical queries involving arbitrary Boolean combinations of attributes. By leveraging the inclusion-exclusion principle, any Boolean query can be converted into Disjunctive Normal Form (DNF). For example, the score for a complex query such as $u \wedge a_1 \vee a_2 \wedge \neg a_3 \vee a_4$ can be rewritten as a sum of scores over several conjunction clauses. Each clause is handled by computing the volume of the intersection of the involved box embeddings. Importantly, the model is trained only on pairwise user–item and attribute–item interactions, yet naturally extends its mechanism to arbitrary, unseen structured logical queries at inference time.

**Time complexity** Each DNF clause requires computing the intersection of multiple boxes. For box embeddings, this is implemented via log-sum-exp (LSE) over coordinate-wise minima and maxima. For a clause involving $k$ variables (user or attributes), the intersection cost is $\mathcal{O}(kD)$, where $D$ is the embedding dimension. To score a full Boolean query with $T$ DNF clauses, the total complexity is $\mathcal{O}(TD)$. In the worst case, where all combinations of $n$ variables appear in disjunction, $T = 2^n$. However, real-world queries are usually structured as conjunctions and simple negations, leading to far fewer terms. Additionally, we parallelize the LSE computations across dimensions and query terms, enabling efficient batched evaluation of logical queries. Our codebase includes these optimizations.

## 4. Experiments

In our experiments, we evaluate all the models on item recommendation across three domains: movies, songs, and restaurants. (4.1). We systematically generate queries of varying complexity from these datasets to evaluate performance on set-theoretic tasks (4.2.1, 4.2.2). We train and select models based on the performance of the traditional personalized item prediction (4.3). Finally, we demonstrate that our set-based representation method is better suited for handling set-theoretic constraints in recommendation tasks (5.1, 5.2).

### 4.1. Dataset

The datasets used in our study must contain two primary components: **Item-User interactions** $\mathcal{D}_U$ and **Item-Attribute interactions** $\mathcal{D}_A$ . We select datasets that offer rich ground truth annotations for both components. We utilize the MovieLens 1M and 20M datasets for personalized movie recommendations (Harper & Konstan, 2015). For the song domain, we employ a subset of the Last-FM dataset, which is the official song tag dataset of the Million Song Dataset (Bertin-Mahieux et al., 2011). In the restaurant domain, we use the NYC-R dataset introduced by (Wang et al., 2018).

We utilize the data curated by Dasgupta et al. (2023) to construct $\mathcal{D}_A$ for the Movielens data. This dataset employs Wikidata (Vrandečić & Krötzsch, 2014) to generate ground truth attribute labels for movies[1]. For the Last-FM dataset, the authors use the Last.fm API ('getTopTags')[2] to create

---

[1] https://github.com/google-research-datasets/genre2movies
[2] https://www.last.fm/

attribute tags. Likewise, the authors in (Wang et al., 2018) crawl restaurant review data from TripAdvisor[3] to curate tags and ratings for restaurants in NYC. The sparsity of $D_A$ and $D_U$ is comparable in the Movielens datasets. In contrast, the Last.fm and NYC-R datasets, designed with tag annotations in mind, exhibit much denser attribute-movie interaction. Thus, the selection of these three datasets not only encompasses diverse domains but also offers varying ground-truth distributions for our experiments.

We use the binarized implicit feedback data (Hu et al., 2008), indicating whether the user or the attribute has been associated with the specific item. To ensure the quality of the data, we retain users/items with 5 or more interactions and attributes with frequency 20 or more in all the datasets. Refer to Table 1 for a detailed description of the dataset statistics.

### 4.2. Dataset Splits & Query Generation

To select models for each method, we train on a dataset split $D_U^{\text{train}}$ & $D_A^{\text{train}}$ while evaluating on a held-out set $D_U^{\text{eval}}$ & $D_A^{\text{eval}}$. However, we use these eval set pairs to construct compositional queries. Simple random sampling or leave-one-out data splits do not ensure a substantial number of these queries. Therefore, we devise a data splitting technique closely linked to query generation, which we discuss next.

#### 4.2.1. PERSONALIZED SIMPLE QUERY

This type of query corresponds to a single attribute for a particular user, *e.g. Bob wants to watch a comedy movie.* More formally, given a user $u$ and an attribute $a$, the query type would be - $u \cap a$. Note that, these simple queries are set-theoretic combinations between the item sets corresponding to the users and the attributes. Let us denote the data corresponding to these queries as $Q_{U \cap A}$.

While constructing the $Q_{U \cap A}$ pairs we need to ensure that - if an item is held out for evaluation for a simple query, the individual user-item and attribute-item pair should belong to the evaluation set as well. More formally, $(u, a, i) \in Q_{U \cap A} \iff (u, i) \in \mathcal{D}_U^{\text{eval}} \wedge (a, i) \in \mathcal{D}_A^{\text{eval}}$. To ensure this train/test isolation, we use the sampling algorithm 1 that takes in $D_U$ and $D_A$ and outputs $Q_{U \cap A}$, $\mathcal{D}_U^{\text{train}}, \mathcal{D}_A^{\text{train}}, \mathcal{D}_U^{\text{eval}}, \mathcal{D}_A^{\text{eval}}$ (Refer to Appendix A.1 for more details). The detailed statistics for the splits are provided in Table 1. Also, the statistics for the $Q_{U \cap A}$ are present in Table 2

#### 4.2.2. PERSONALIZED COMPLEX QUERY

The set-theoretic compositions that we consider here are the intersection and negation of attributes for a particular user. Given a user $u$ and attributes $a_1$ and $a_2$, we consider the query types- $u \cap a_1 \cap a_2$ and $u \cap a_1 \cap \neg a_2$, e.g, *Bob want to watch an Action Comedy movie, Alice want to watch a Children but not Monster movie.* Creating meaningful attribute compositions requires careful consideration, as not all combinations make sense. For instance, 'Sci-Fi' & 'Documentary' might not be a meaningful combination, whereas 'Sci-Fi' & 'Time-Travel' is. Similarly, 'Sci-Fi' ¬ 'Fiction' doesn't make sense, but 'Fiction' ¬ 'Sci-Fi' does. Sometimes, even if the intersection is valid, it could be trivial and non-interesting, e.g., 'Fiction' & 'Sci-Fi'.

Intuitively, for two attributes $a_1$ & $a_2$, their intersection is interesting if $|a_1 \cap a_2|$ is greater than combining any two random items set. Also, for their intersection to be non-trivial the size of the intersection $|a_1 \cap a_2|$ must be less than the individual sizes of the attributes i.e., $\alpha|a_1|$ and $\alpha|a_2|$. Here,$|.|$ denotes the size of the item set corresponding to the attributes. $\alpha \in [0, 1]$ is a design parameter, dedicated after manual inspection of the quality of the item sets for the combinations . In case of difference queries such as $a_1 \cap \neg a_2$, we consider $\neg a_2$ to be the second attribute and carry out the same filtering strategy as done for the intersection queries. We denote the set of non-trivial and viable attribute pairs for the intersection to be $\mathcal{A}_\cap = \{(a_1, a_2) | |a_1 \cap a_2| > \epsilon, |a_1 \cap a_2| < \alpha|a_1|, |a_1 \cap a_2| < \alpha|a_2|\}$, and for the difference to be $\mathcal{A}_\setminus = \{(a_1, a_2) | |a_1 \cap \neg a_2| > \epsilon, |a_1 \cap \neg a_2| < \alpha|a_1|, |a_1 \cap \neg a_2| < \alpha|\neg a_2|\}$. Using the above formulation, we generate the test set for the personalized complex queries $Q_{U \cap A_1 \cap A_2}$ and $Q_{U \cap A_1 \cap \neg A_2}$ using algorithm 2. Please refer to Table 2 for the detailed statistics. The link for the dataset is available at https://github.com/ssdasgupta/set-based-collaborative-filtering.

### 4.3. Training Details & Evaluation Criteria

We train all the methods on users and attributes jointly using $\mathcal{D}^{\text{train}} = \mathcal{D}_U^{\text{train}} \cup \mathcal{D}_A^{\text{train}}$. We use dimensions $d = 128$ for vector-based models, and $d = 64$ for box models so that the number of parameters per user, attribute, and movie is equal.[4] We perform extensive hyperparameter tuning for the learning rate, batch size, volume and intersection temperature of boxes, loss combination constant, etc. Please refer to the Appendix A.2 for details. We follow the standard sampled evaluation procedure described in Rendle et al. (2020), only for model selection purpose. For each user-item tuple $(u, m)$ in $\mathcal{D}_U^{\text{eval}}$, the model ranks $m$ amongst a set of items consisting of the $m$ together with 100 other true negative items w.r.t the user. Then we report on two different evaluation metrics namely Hit Ratio@$k$ (HR@$k$) and NDCG. (a) HitRatio@$k$: If the rank of $m$ is less than or equals to $k$ then the value of HR@$k$ is 1 or 0 otherwise. (2) NDCG: if $r$ is the rank of $m$, then $1/\log(r + 1)$ is the NDCG.

---

[3] https://www.tripadvisor.com

[4] Recall that box embeddings are parameterized with two vectors, one for each min and max coordinate.

Table 1: Dataset Statistics, the Item-User interaction $\mathcal{D}_U$ & the Item-Attribute interaction $\mathcal{D}_A$.
The Train/Test split is created using algorithm 1 to test set-theoretic generalization.

| Dataset | #Users | #Items | #Attributes | #Train $\mathcal{D}_U$ | #Eval $\mathcal{D}_U$ | #Train $\mathcal{D}_A$ | #Eval $\mathcal{D}_A$ |
|---|---|---|---|---|---|---|---|
| Last-FM | 1,872 | 2417 | 490 | 60,497 | 8,857 | 34,374 | 4,240 |
| NYC-R | 9,597 | 3764 | 579 | 82,734 | 8,502 | 34,908 | 4,376 |
| MovieLens 1M | 6,040 | 3,705 | 57 | 963,554 | 36,655 | 10,273 | 1,545 |
| MovieLens-20M | 138,493 | 26,744 | 95 | 19,722,646 | 277,617 | 80,178 | 1,734 |

The model is selected based on the best-performing model on NDCG for the item prediction over the user-item validation set $\mathcal{D}_U^{\text{eval}}$, with the best-performing checkpoint saved for further evaluation on compositional queries. We follow the same evaluation protocol for the compositional queries as well, except, we rank $m$ amongst all items in the vocabulary rather than a sampled subset.

### 4.4. Baselines

The recommendation systems literature offers a wide range of methods that represent users, and items in $\mathbb{R}^d$. These methods then propose a compatibility score function between the user and item, $\phi : \mathbb{R}^d \times \mathbb{R}^d \to \mathbb{R}$. A common and effective choice for $\phi$ is the dot product, which underpins matrix factorization (Rendle et al., 2020; Koren & Bell, 2015). To capture more complex interactions among users, items, and attributes, (He et al., 2017) extend matrix factorization by replacing the dot product with a neural network-based similarity function. This method, called Neural Matrix Factorization (NEUMF), combines the dot product with an MLP. Similarly, (He et al., 2020) propose LightGCN (LGCN) to captures the user, items, and attribute interaction using Graph Convolution Network (Kipf & Welling, 2017) over a joint graph of user-item-attribute. We use MF, and, NEUMF LGCN as our baselines.

For a personalized query, be it simple or complex, we need to devise a method to combine the individual scores of the user and the attributes involved in the query. In this work, we compare three approaches to obtain an aggregated score:

1. FILTER: In this approach, we retrieve a list of items corresponding to the attributes based on the scores provided by the embedding models. The list is generated by thresholding the scores, where the threshold is optimized by minimizing the F1 score between the training data and predicted scores. We refer to the methods using this aggregation technique as BOX-FILTER for box embeddings and MF-FILTER, NEUMF-FILTER, LGCN-FILTERfor vector-based methods.

2. PRODUCT: In this method, the compositional score is computed by multiplying the scores for the individual queries. For vector-based embeddings, the scores for

Table 2: Compositional Query Statistics

| Dataset | Personalized Simple Query $u \cap a$ | Personalized Complex Query | |
|---|---|---|---|
| | | $u \cap a_1 \cap a_2$ | $u \cap a_1 \cap \neg a_2$ |
| Last-FM | 9,867 | 45,142 | 10,814 |
| NYC-R | 9,482 | 7,460 | 2,369 |
| ML-1M | 21,392 | 51,299 | 37,769 |
| ML-20M | 35,368 | 42,355 | 47,374 |

each movie related to a user or attribute are normalized using the *sigmoid* function. For box embeddings, the energy function is normalized by conditioning on the movie box volume (see Section 3.3). The score for negation is calculated by subtracting the normalized score from 1. The three methods using this technique are referred to as BOX-PRODUCT, MF-PRODUCT, NEUMF-PRODUCT, and LGCN-PRODUCT.

3. GEOMETRIC: This approach leverages the geometry of the embedding space. For vector-based embeddings, learned through Matrix Factorization, addition, and subtraction are often used for query composition (Mikolov et al., 2013). Box embeddings, on the other hand, naturally represent intersection operations, allowing us to compute scores for any set-theoretic combination using box intersection and inclusion-exclusion principles. We refer to these methods as BOX-GEOMETRIC and MF-GEOMETRIC.

## 5. Results

After conducting an extensive hyper-parameter search on $D_U^{\text{eval}}$, we select the top-performing model for each method based on NDCG scores (see Table 6 in the Appendix for the model selection details). This ensures that the chosen model is optimal for set-theoretic query inference, with the following performance results.

### 5.1. Set-Theoretic Generalization

We test the selected models for each method with the curated set-theoretic personalized queries (Detailed stats for the

Table 3: Hit Rate(%)↑ on Set-theoretic queries for datasets Last-FM, MovieLens 1M, NYC-R.

| Methods | $U \cap A$ | | | $U \cap A_1 \cap A_2$ | | | $U \cap A_1 \cap \neg A_2$ | | |
|---|---|---|---|---|---|---|---|---|---|
| | h@10 | h@20 | h@50 | h@10 | h@20 | h@50 | h@10 | h@20 | h@50 |
| LAST-FM | | | | | | | | | |
| MF-FILTER | 14.8 | 25.1 | 37.4 | 26.8 | 46.8 | 62.8 | 15.2 | 24.4 | 35.5 |
| MF-PRODUCT | 9.0 | 21.7 | 48.0 | 14.3 | 36.8 | 73.2 | 4.8 | 14.8 | 43.4 |
| MF-GEOMETRIC | 6.1 | 12.2 | 29.7 | 3.4 | 7.6 | 27.5 | 1.7 | 4.8 | 15.9 |
| NEUMF-FILTER | 13.5 | 21.9 | 32.3 | 20.0 | 19.6 | 55.7 | 11.3 | 18.8 | 28.7 |
| NEUMF-PRODUCT | 13.6 | 25.6 | 47.6 | 19.5 | 35.7 | 63.3 | 9.0 | 16.8 | 40.5 |
| LGCN-FILTER | 20.4 | 28.5 | 39.1 | 42.4 | 54.2 | 67.4 | 15.8 | 21.5 | 27.6 |
| LGCN-PRODUCT | 20.5 | 31.0 | 48.6 | 43.8 | 58.0 | 80.7 | 0.8 | 1.3 | 3.5 |
| BOX-FILTER | 22.9 | 31.5 | 39.0 | 32.7 | 46.5 | 55.9 | 22.0 | 32.1 | 40.3 |
| BOX-PRODUCT | 27.9 | 44.5 | 68.0 | 38.2 | 57.7 | 82.7 | 17.8 | 32.4 | 60.3 |
| BOX-GEOMETRIC | 28.3 | 44.8 | 68.3 | 38.8 | 58.3 | 83.1 | 17.5 | 32.5 | 60.0 |
| MOVIELENS-1M | | | | | | | | | |
| MF-FILTER | 5.0 | 10.2 | 22.3 | 11.4 | 17.9 | 27.5 | 4.7 | 9.8 | 22.5 |
| MF-PRODUCT | 4.3 | 8.5 | 20.4 | 5.1 | 10.6 | 26.1 | 3.4 | 7.3 | 19.3 |
| MF-GEOMETRIC | 0.4 | 0.9 | 3.0 | 0.1 | 0.2 | 0.8 | 0.5 | 1.0 | 2.7 |
| NEUMF-FILTER | 9.3 | 15.5 | 28.5 | 13.3 | 21.5 | 35.9 | 8.8 | 14.7 | 26.7 |
| NEUMF-PRODUCT | 10.3 | 16.8 | 31.4 | 15.3 | 24.5 | 43.5 | 5.7 | 9.7 | 20.2 |
| LGCN-FILTER | 8.2 | 12.3 | 20.9 | 11.4 | 15.6 | 24.0 | 9.9 | 13.8 | 21.9 |
| LGCN-PRODUCT | 5.9 | 9.0 | 14.9 | 7.6 | 11.7 | 20.1 | 5.5 | 8.6 | 14.1 |
| BOX-FILTER | 11.7 | 19.1 | 32.3 | 14.5 | 20.5 | 28.6 | 11.4 | 19.5 | 34.0 |
| BOX-PRODUCT | 9.95 | 16.7 | 31.5 | 10.6 | 17.8 | 34.2 | 8.9 | 15.1 | 29.4 |
| BOX-GEOMETRIC | 11.0 | 18.3 | 34.2 | 16.9 | 26.6 | 46.1 | 8.6 | 15.2 | 31.0 |
| NYC-R | | | | | | | | | |
| MF-FILTER | 1.4 | 2.4 | 4.6 | 2.7 | 4.8 | 8.0 | 2.1 | 3.5 | 6.3 |
| MF-PRODUCT | 1.1 | 2.9 | 8.6 | 3.7 | 8.2 | 23.3 | 8.9 | 13.1 | 17.6 |
| MF-GEOMETRIC | 0.5 | 1.5 | 4.3 | 0.2 | 0.8 | 3.5 | 0.5 | 1.2 | 3.7 |
| NEUMF-FILTER | 3.8 | 5.6 | 9.2 | 2.5 | 3.2 | 4.5 | 4.2 | 6.3 | 10.8 |
| NEUMF-PRODUCT | 4.6 | 7.3 | 13.7 | 6.6 | 11.2 | 20.8 | 2.7 | 5.2 | 11.2 |
| LGCN-FILTER | 4.8 | 7.8 | 17.2 | 12.7 | 16.9 | 21.8 | 5.4 | 8.6 | 16.4 |
| LGCN-PRODUCT | 5.0 | 8.7 | 18.1 | 12.1 | 17.6 | 35.1 | 4.9 | 8.0 | 13.2 |
| BOX-FILTER | 4.9 | 7.8 | 13.4 | 9.9 | 13.5 | 20.4 | 4.4 | 7.1 | 12.5 |
| BOX-PRODUCT | 5.0 | 8.9 | 17.9 | 10.9 | 19.5 | 37.3 | 5.3 | 9.1 | 18.8 |
| BOX-GEOMETRIC | 4.9 | 8.7 | 17.6 | 12.2 | 21.5 | 39.2 | 5.5 | 9.2 | 19.2 |

Table 4: *Generalization Spectrum Gap* for PERSONALIZED COMPLEX QUERY $U \cap A_1 \cap A_2$

| Methods | Hit Rate @50 ↑ | | | | *Spectrum Gap ↓* |
|---|---|---|---|---|---|
| | Weakest (W) | Weak-User (W-U) | Weak-Attribute (W-A) | Set-Theoretic (S) | (W − S) / W |
| MF-FILTER | 55.2 | 41.9 | 30.5 | 27.5 | 50.2% |
| MF-PRODUCT | 67.4 | 38.5 | 39.3 | 26.1 | 61.2 % |
| MF-GEOMETRIC | 18.5 | 12.9 | 1.8 | 0.8 | 95.6% |
| NEUMF-FILTER | 48.4 | 33.1 | 40.4 | 35.9 | 38.5% |
| NEUMF-PRODUCT | 67.8 | 48.7 | 40.6 | 43.5 | 35.9% |
| BOX-FILTER | 52.7 | 44.5 | 30.3 | 28.5 | 45.9% |
| BOX-PRODUCT | 64.6 | 52.8 | 39.0 | 34.2 | 47.1% |
| BOX-GEOMETRIC | 62.6 | 53.3 | 50.1 | 46.1 | 26.4% |

queries in Table 2). We report the ranking performance in terms of Hit Rates at 10, 20, and 50. Please refer to 3 for the results.

The Box Embedding-based method outperforms vector-based methods by a significant margin, showing on average 30% improvement when comparing the aggregated HR@50 performance of the best vector model (MF-PRODUCT/NEUMF-PRODUCT/LGCN-FILTER) to the box model (BOX-GEOMETRIC) across all the three different domains.

The $U \cap A_1 \cap A_2$ query is the most challenging, as it requires accuracy in all three individual queries. For this difficult query, BOX-GEOMETRIC shows the largest performance gap compared to other methods. Additionally, using vector addition and subtraction as geometric proxies for intersection and difference performs significantly worse than all other vector-based methods, while geometric operations in the box embedding space outperform even other box embedding methods. This validates the set-theoretic inductive bias of box embeddings and confirms that geometric operations in this space provide valid set-theoretic operations, unlike vectors.

The FILTER aggregation technique performs similarly to or better than other methods only for Hits@10. However,

as $k$ increases, its performance declines across all model types (Box, MF, NeuMF) and datasets. This observation highlights the limitation of a fixed threshold filter and advocates smoother aggregation techniques like PRODUCT and GEOMETRIC.

### 5.2. Spectrum of Generalization

The query generation process (refer Section 4.2.1) ensures that for the target item $m$ corresponding to a query involving user $u$ and attribute $a$, the pair $(u, m)$ and $(a, m)$ must not be in the training set $(u, m) \notin \mathcal{D}_U^{train}$ and $(a, m) \notin \mathcal{D}_A^{train}$. The set-theoretic evaluation weakens when such pairs are added back to the training set. There are three different weakening settings applicable here, which we refer to as a spectrum – WEAKEST GENERALIZATION ($(u, m) \in \mathcal{D}_U^{train}$ and $(a, m) \in \mathcal{D}_A^{train}$), WEAK GENERALIZATION-USER ($(u, m) \in \mathcal{D}_U^{eval}$ and $(a, m) \notin \mathcal{D}_A^{train}$), WEAK GENERALIZATION-ATTRIBUTE ($(u, m) \notin \mathcal{D}_U^{train}$ and $(a, m) \in \mathcal{D}_A^{eval}$). We report HitRate@50 performance on query type $U \cap A_1 \cap A_2$ for the MovieLens-1M dataset in Table 4 (More query types in Appendix - Table 9, 8).

The weaker the generalization setting the easier it is for the models to achieve higher performance on the test set. Indeed, we observe that this is true across all the methods w.r.t each of the aggregation settings, validating the correctness of the trained models.

However, we are interested in observing the performance gap when we go from the weakest to the strongest set-theoretic generalization. We refer to the percentage gap *Generalization Spectrum Gap* (hr(Weakest) - hr(Set-theoretic) / hr(Weakest) %). From Table 4 we observe that the best-performing box model BOX-GEOMETRIC achieves the best *Generalization Spectrum Gap* for HR@50.

## 6. Related Work

### 6.1. Box Embeddings

Some of the recent works have tried to incorporate box embeddings in a recommendation systems setup.Xu et al. (2024); Wu et al. (2024); Zhang et al. (2021) use the side-length of the box embeddings as a preference range to obtain

diverse set recommendations for users, Mei et al. (2022a) utilizes the axis parallel nature of the box embeddings for faster retrieval. Sun et al. (2020a;b); Ren et al. (2020) are some of the recent works that focus on logical query over knowledge bases (KB). However, in this work, we frame collaborative filtering as a set-theoretic matrix completion problem, which helps us to achieve better generalization for the composition of personalized queries.

### 6.2. Set-based queries in Search and group recommendation systems.

While set-theoretic queries are commonplace in search, popular question-answering (QA) benchmarks often do not include them. We found QUEST (Malaviya et al., 2023) the most closely related study, introducing a benchmark for entity-seeking queries with implicit set-based semantics. However, QUEST does not focus on explicit constraints or personalization, which are central to our work.

### 6.3. Context Aware Recommendation

The concept of context-aware recommendation, as introduced in (Adomavicius et al., 2011), provides a general framework where "context" is broadly defined as any auxiliary information. This framework emphasizes that user preferences for items can vary based on the context in which interactions occur, reflecting a user-centric view of contextual information.

Building on this foundation, recent works have explored specific instances of context-aware recommendation, such as "attribute-aware recommendation." These approaches often leverage item or user attributes as contextual information to address various goals, including improving user profiling (Adomavicius et al., 2011), predicting missing item attributes (Wu et al., 2020; Chen et al., 2022), enhancing recommendations for cold-start scenarios(Deldjoo et al., 2019), or providing attribute-based explanations for recommendations (Xian et al., 2021).

Our work differs significantly in its focus and objectives. we term "attribute-constrained recommendation," which involves generating recommendations explicitly constrained by logical combinations of attributes. Unlike attribute-aware approaches, which aim to improve recommendation quality by incorporating attribute information as auxiliary data, our work directly targets the task of satisfying explicit attribute-based constraints posed by users.

### 6.4. Compositional Queries with Vector Embeddings

It is common in machine learning to represent discrete entities such as items or attributes by vectors (Bengio et al., 2013) and to learn them by fitting the training data. Besides semantic similarity, some have claimed that learned vectors have compositional properties through vector arithmetic, for example in the empirical analysis of word2vec (Mikolov et al., 2013) and GLOVE (Pennington et al., 2014), and some theoretical analysis (Levy & Goldberg, 2014; Arora et al., 2018). However, anecdotally, many have found that the compositional behavior of vectors is far from reliable (Rogers et al., 2017). Our paper provides a comprehensive evaluation of vector embeddings on compositional queries and compares the results to a region-based alternative.

## 7. Conclusion & Future Work

In this work we presented the task of personalized recommendation with set-theoretic queries. We discussed how this problem can be viewed as set-theoretic matrix completion, and why the common approach of logistic matrix factorization is not aligned with the set-theoretic operations we wish to perform at inference time. We observed substantial improvements over the vector/neural baselines when using box embeddings as the representation, validating our intuition regarding the necessary set-theoretic bias. Our empirical results confirm that box embeddings are ideally suited to the task of recommendation with set-theoretic queries.

In real-world recommendation systems — such as streaming platforms, e-commerce sites, or travel services — free-text queries (e.g., "funny action movies without clowns") are typically mapped to a curated set of item tags (e.g., genre, theme, metadata) via a natural language understanding (NLU) module. Our model operates downstream of this step, assuming a structured query (e.g., Action ∧ Comedy ∧ ¬ Clowns) has already been derived. While our current focus is on the execution of structured set-theoretic queries, future work could explore tighter integration with front-end NLU systems. Large language models (e.g., GPT-4o) have demonstrated strong performance in parsing natural language constraints, and we view such models as complementary: performing query parsing and attribute identification, while our method serves as a reliable and efficient back-end executor for the resulting set-theoretic logic. Bridging the gap between these stages offers a promising direction for building end-to-end systems that are both expressive and controllable.

As noted in Section 3.3, our model supports efficient evaluation of complex queries. We construct a benchmark of semantically plausible queries using statistical heuristics and manual filtering (Section 4.2.2), ensuring realistic and diverse combinations. While this allows us to test compositional generalization, curating a benchmark of natural user-issued queries remains an important direction for future work.

## Acknowledgments

The authors would like to thank the members of the Information and Extraction Synthesis Laboratory (IESL) at UMass Amherst, Steffen Rendle, and Li Zhang, for helpful discussions. This work was supported by IBM Research AI through the AI Horizons Network and National Science Foundation (NSF) under the Grant Numbers IIS-2106391. The views and conclusions contained herein are those of the authors and should not be interpreted as necessarily representing the official policies or endorsements, either expressed or implied, of IBM or NSF.

## Impact Statement

This paper aims to advance the field of Machine Learning by introducing a geometric approach to personalized recommendation under set-theoretic constraints. Our primary contribution is methodological, focusing on improving representation learning for structured preference modeling. While recommendation systems have broad societal reach and their deployment may influence user behavior, fairness, or exposure to information, this work does not involve direct deployment or sensitive user data. As such, we do not identify any immediate or domain-specific societal risks associated with this research. Nonetheless, we acknowledge the importance of responsible use and encourage future applications of our method to consider fairness, transparency, and user control as core design considerations.

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

Table 6: Test NDCG on $D_U^{\text{eval}}$ for selected models.

| Dataset | MF | NeuMF | Lgcn | Box |
|---|---|---|---|---|
| Last-FM | 0.51 | 0.52 | 0.56 | 0.65 |
| NYC-R | 0.31 | 0.33 | 0.37 | 0.39 |
| ML-1M | 0.51 | 0.53 | 0.55 | 0.58 |
| ML-20M | 0.71 | 0.70 | 0.72 | 0.73 |

# A. Experiment Details

## A.1. Data Splits & Query Generation

---

**Algorithm 1** PERSONALISED SIMPLE QUERY $(u \cap a)$ generation algorithm $u \cap a$

---

1: Let the set of users, attributes, and movies be $\mathcal{U}, \mathcal{A}, \mathcal{M}$
2: Marginal probability of an attribute $a$ in $A$, $P(a) = \sum_m A_{a,m} / \sum_{a'} \sum_m A_{a',m}$
3: Marginal probability of an user $u$ in $U$, $P(u) = \sum_m U_{u,m} / \sum_{u'} \sum_m U_{u',m}$
4: Marginal probability of an movie $m$ in $U$, $P(m) = \sum_u U_{u,m} / \sum_u \sum_{m'} U_{u,m'}$
5: Let $U$ be the User $\times$ Item matrix and $A$ be the Attribute $\times$ Item matrix.
6: $U^{Train} \leftarrow U, A^{Train} \leftarrow A$
7: $U^{Eval} \leftarrow \mathbf{0}, A^{Eval} \leftarrow \mathbf{0}$
8: Set of simple personalized queries, $Q_{U \cap A} \leftarrow \phi$
9: **while** $|Q_{U \cap A}| <$ MAX SAMPLE SIZE **do**
10:     Sample an attribute $a$ from $\mathcal{A}$ according to $P(a)$.
11:     Sample a movie $m$ from for the attribute $a$, i.e., Sample from $\{m'|A_{a,m'} = 1\}$, according to $P(m)$
12:     Sample a user $u$ from who has rated movie $m$, i.e., Sample from $\{u'|U_{m,u'} = 1\}$, according to $P(u)$
13:     $U_{u,m}^{Train} = 0, A_{a,m}^{Train} = 0, U_{u,m}^{Eval} = 1, A_{a,m}^{Eval} = 1$
14:     $Q_{U \cap A}$.INSERT$((u, a, m))$
15: **end while**

---

## A.2. Training Details

Table 5: Hyper Parameter range for all the dataset. We run 100 runs for both models and select the best model on User-Movie validation set NDCG metric

| Hyperparameters | Range Box | Best Value Box | Range Vector | Best Value Vector |
|---|---|---|---|---|
| Embedding dim | 64 | 64 | 128 | 128 |
| Learning Rate | 1e-1, 1e-2, 1e-3, 1e-4, 1e-5 | 0.001 | 1e-1, 1e-2, 1e-3, 1e-4, 1e-5 | 0.001 |
| Batch Size | 64, 128, 256, 512, 1024 | 128 | 64, 128, 256, 512, 1024 | 128 |
| # Negatives | 1, 5, 10, 20 | 20 | 1, 5, 10, 20 | 5 |
| Intersection Temp | 10, 2, 1, 1e-1, 1e-2, 1e-3, 1e-5 | 2.0 | - | - |
| Volume Temp | 10, 5, 1, 0.1, 0.01, 0.001 | 0.01 | - | - |
| Attribute Loss const | 0.1, 0.3, 0.5, 0.7, 0.9 | 0.7 | 0.1, 0.3, 0.5, 0.7, 0.9 | 0.5 |

Hyperparameters are reported in Table 5. Best parameter values are reported for Box Embeddings and MF method.

---

**Algorithm 2** PERSONALISED COMPLEX QUERY Generation Algorithm

---

1: Compositional Query sets $Q_{U \cap A_1 \cap A_2}, Q_{U \cap A_1 \cap \neg A_2}$
2: Non-Trivial attribute combination set $\mathcal{A}_\circ$
3: **for** each user-movie tuple in Eval set, i.e., $(u, m) \in \{(u,m)|U_{u,m}^{Eval} = 1\}$ **do**
4:     **for** each pair of attributes $(a_1, a_2) \in \{(a_1, a_2)|A_{a_1,m}^{Eval} = 1 \text{ and } A_{a_2,m}^{Eval} = 1\}$ **do**
5:         **if** the pair is viable and non-trivial, i.e., $(a_1, a_2) \in \mathcal{A}_\cap$ **then**
6:             $Q_{U \cap A_1 \cap A_2}$.INSERT$((u, a_1, a_2, m))$
7:         **end if**
8:     **end for**
9:     **for** each pair of attributes $(a_1, a_2) \in \{(a_1, a_2)|A_{a_1,m}^{Eval} = 1 \text{ and } A_{a_2,m} = 0\}$ **do**
10:         **if** the pair is viable and non-trivial, i.e., $(a_1, a_2) \in \mathcal{A}_\setminus$ **then**
11:             $Q_{U \cap A_1 \cap \neg A_2}$.INSERT$((u, a_1, a_2, m))$
12:         **end if**
13:     **end for**
14: **end for**

---

## A.3. Model Selection

## A.4. Set-Theoretic Generalization

Table 7: Hit Rate(%)↑ for Set-theoretic queries for dataset ML-20M.

| Methods | $U \cap A$ | | | $U \cap A_1 \cap A_2$ | | | $U \cap A_1 \cap \neg A_2$ | | |
|---|---|---|---|---|---|---|---|---|---|
| | h@10 | h@20 | h@50 | h@10 | h@20 | h@50 | h@10 | h@20 | h@50 |
| MF-FILTER | 4.6 | 8.1 | 16.1 | 0.4 | 1.0 | 2.9 | 3.7 | 6.6 | 13.7 |
| MF-PRODUCT | 4.1 | 7.5 | 15.6 | 3.3 | 6.6 | 16.4 | 2.7 | 5.1 | 11.4 |
| MF-GEOMETRIC | 0.1 | 0.3 | 0.6 | 0.0 | 0.0 | 0.0 | 0.3 | 0.6 | 1.4 |
| NEUMF-FILTER | 4.6 | 8.2 | 16.1 | 1.1 | 5.6 | 6.4 | 4.9 | 7.3 | 13.9 |
| NEUMF-PRODUCT | 4.6 | 8.2 | 16.1 | 4.1 | 8.5 | 22.1 | 4.3 | 6.9 | 12.0 |
| BOX-FILTER | 4.6 | 8.1 | 16.1 | 11.0 | 21.8 | 42.3 | 4.6 | 7.7 | 16.3 |
| BOX-PRODUCT | 4.5 | 8.2 | 16.1 | 11.1 | 21.8 | 42.5 | 4.3 | 7.1 | 15.1 |
| BOX-GEOMETRIC | 4.5 | 8.1 | 16.2 | 11.0 | 21.8 | 42.4 | **6.4** | **12.8** | **25.9** |

## A.5. Spectrum of Weak Generalization

Table 8: The spectrum of generalization for SIMPLE PERSONALIZED query $U \cap A$. W: WEAKEST GENERALIZATION, W-U: WEAK GENERALIZATION-USER, W-A: WEAK GENERALIZATION-ATTRIBUTE, S: SET THEORETIC GENERALIZATION

| Methods | Hit Rate @10 | Hit Rate @ 20 | Hit Rate @ 50 |
|---|---|---|---|
| | W \| W-U \| W-A \| S | W \| W-U \| W-A \| S | W \| W-U \| W-A \| S |
| MF-FILTER | 24.7 \| 6.7 \| 13.0 \| 5.0 | 36.3 \| 13.3 \| 20.7 \| 10.2 | 54.2 \| 30.1 \| 33.3 \| 22.3 |
| MF-PRODUCT | 23.3 \| 15.7 \| 13.1 \| 4.3 | 35.0 \| 10.8 \| 21.4 \| 8.5 | 54.7 \| 24.2 \| 38.8 \| 20.4 |
| MF-GEOMETRIC | 4.9 \| 0.9 \| 1.8 \| 0.4 | 7.9 \| 1.7 \| 3.3 \| 0.9 | 15.1 \| 4.5 \| 7.4 \| 3.0 |
| BOX-FILTER | 24.1 \| 13.0 \| 16.4 \| 11.7 | 34.5 \| 22.3 \| 24.6 \| 19.1 | 50.5 \| 40.5 \| 37.6 \| 32.3 |
| BOX-PRODUCT | 25.2 \| 13.6 \| 13.9 \| 10.0 | 35.2 \| 21.5 \| 21.9 \| 16.7 | 52.2 \| 38.4 \| 38.3 \| 31.5 |
| BOX-GEOMETRIC | 25.4 \| 14.7 \| 14.8 \| 11.0 | 35.6 \| 23.3 \| 23.5 \| 18.3 | **52.2** \| **40.8** \| **40.5** \| **34.1** |

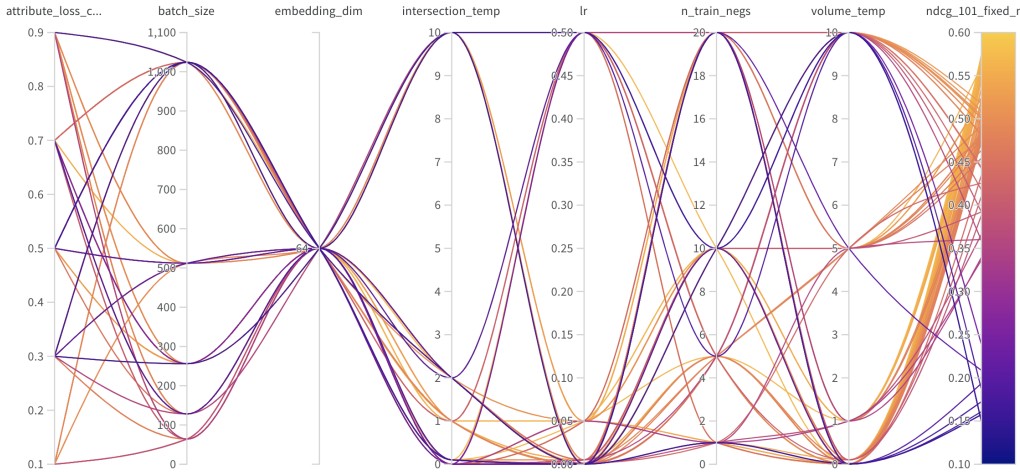

Figure 3: Parallel Co-ordinate plot for different hyperparameters vs model performance. Lighter the color, better the model's performance.

Table 9: The spectrum of generalization for COMPLEX PERSONALIZED QUERY $U \cap A_1 \cap \neg A_2$. W: WEAKEST GENERALIZATION, W-U: WEAK GENERALIZATION-USER, W-A: WEAK GENERALIZATION-ATTRIBUTE, S: SET THEORETIC GENERALIZATION

| Methods | Hit Rate @10 | Hit Rate @ 20 | Hit Rate @ 50 |
|---|---|---|---|
| | W \| W-U \| W-A \| S | W \| W-U \| W-A \| S | W \| W-U \| W-A \| S |
| MF-FILTER | 25.5 \| 13.0 \| 12.4 \| 4.7 | 34.9 \| 14.1 \| 19.5 \| 9.8 | 54.7 \| 29.5 \| 37.1 \| 22.5 |
| MF-PRODUCT | 23.5 \| 7.0 \| 10.4 \| 3.4 | 34.9 \| 12.8 \| 18.0 \| 7.3 | 54.5 \| 27.5 \| 35.0 \| 19.3 |
| MF-GEOMETRIC | 5.2 \| 2.0 \| 1.7 \| 0.5 | 8.8 \| 3.5 \| 1.9 \| 1.0 | 17.4 \| 8.8 \| 6.5 \| 2.7 |
| BOX-FILTER | 24.1 \| 15.3 \| 15.0 \| 11.4 | 35.5 \| 22.7 \| 21.1 \| 19.5 | **54.1** \| **39.2** \| **37.3** \| **34.0** |
| BOX-PRODUCT | 21.1 \| 13.7 \| 12.0 \| 8.9 | 30.5 \| 21.7 \| 19.3 \| 15.2 | 47.4 \| 38.0 \| 35.0 \| 29.4 |
| BOX-GEOMETRIC | 21.1 \| 13.2 \| 10.8 \| 8.6 | 30.4 \| 20.8 \| 17.7 \| 15.1 | **47.3** \| **36.6** \| **33.2** \| **31.0** |

Table 10: The spectrum of generalization for COMPLEX PERSONALIZED QUERY $U \cap A_1 \cap A_2$. W: WEAKEST GENERALIZATION, W-U: WEAK GENERALIZATION-USER, W-A: WEAK GENERALIZATION-ATTRIBUTE, S: SET THEORETIC GENERALIZATION

| Methods | Hit Rate @10 | Hit Rate @ 20 | Hit Rate @ 50 |
|---|---|---|---|
| | W \| W-U \| W-A \| S | W \| W-U \| W-A \| S | W \| W-U \| W-A \| S |
| MF-FILTER | 35.3 \| 17.6 \| 16.9 \| 11.4 | 45.0 \| 27.3 \| 23.3 \| 17.9 | 55.2 \| 41.9 \| 30.5 \| 27.5 |
| MF-PRODUCT | 34.0 \| 11.0 \| 11.6 \| 5.1 | 47.3 \| 19.6 \| 20.1 \| 10.6 | 67.4 \| 38.5 \| 39.3 \| 26.1 |
| MF-GEOMETRIC | 6.13 \| 3.1 \| 0.3 \| 0.1 | 9.90 \| 5.8 \| 0.6 \| 0.2 | 18.5 \| 12.9 \| 1.8 \| 0.8 |
| BOX-FILTER | 30.8 \| 21.5 \| 17.3 \| 14.5 | 41.1 \| 31.2 \| 23.3 \| 20.5 | 52.7 \| 44.5 \| 30.3 \| 28.5 |
| BOX-PRODUCT | 35.4 \| 23.8 \| 13.4 \| 10.6 | 47.0 \| 34.5 \| 21.7 \| 17.8 | 64.6 \| 52.8 \| 39.0 \| 34.2 |
| BOX-GEOMETRIC | 34.6 \| 25.2 \| 20.0 \| 16.8 | 45.7 \| 35.7 \| 30.5 \| 26.6 | **62.6** \| **53.3** \| **50.1** \| **46.1** |

The BOX-GEOMETRIC achieves the best *Generalization Spectrum Gap* for all types of queries.

# B. Error Compounding Analysis

We further perform more granular analysis amongst the BOX based methods with complex query type $U \cap A_1 \cap A_2$.

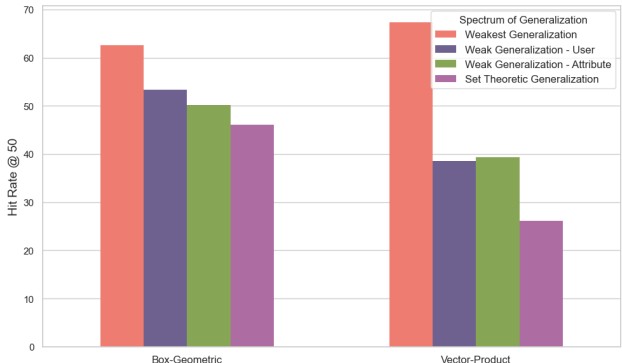

Figure 4: Weak Generalization Illustration

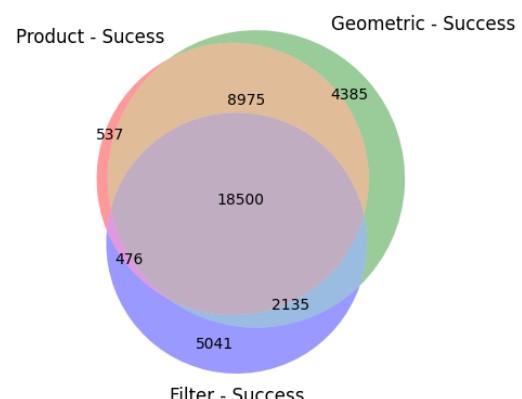

Figure 5: Relationships of correct answers by the three box models on $u \wedge a_1 \wedge a_2$ queries.

As claimed in our initial hypothesis, the FILTER method suffers from error compounding. If the target movie $m$ is

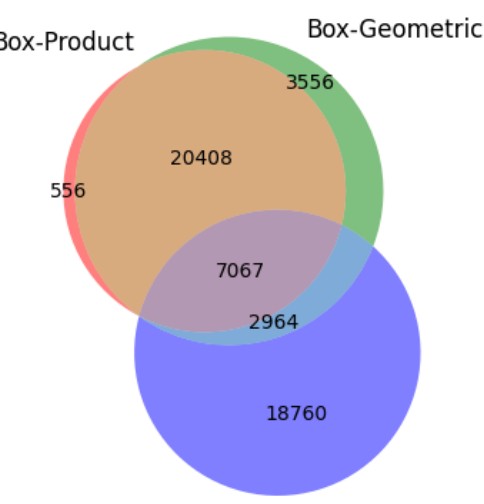

Figure 6: The Geometric method subsumes the benefit of the product in compounding error.

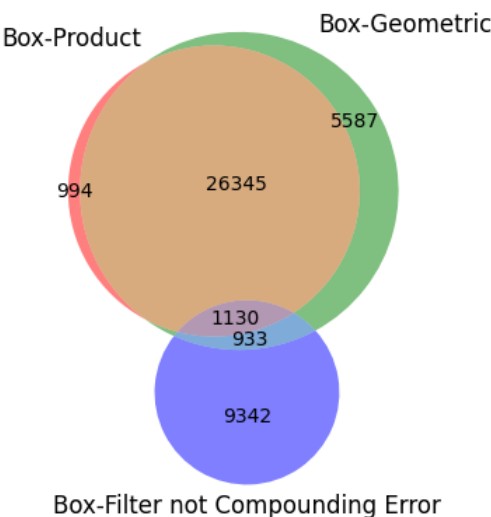

Figure 7: The effect is less for the non-compounding error.

in the model's prediction list for $A_1$ but not for $A_2$ or the other way round, we denote this error as *compounding error*. In figure 6, out of the compounding errors, $34\%$ is solved by the BOX-GEOMETRIC method and $26\%$ by the BOX-PRODUCT method. However, in figure 7, for the error that is not due to compounding (where the model gets both $A_1$ and $A_2$ prediction wrong), only $18\%$ are corrected by the BOX-GEOMETRIC method and a mere $10\%$ of them are corrected by BOX-PRODUCT. Refer to figure 5 6 7 for details. This demonstrates that the BOX-GEOMETRIC significantly contributes to the correction of error compounding.

## C. Time Efficiency analysis

Table 11: Training time (*mm:ss*) for a single epoch are measured for different batch sizes with 5 negative samples on Movielens-1M dataset. Experiments are conducted on Nvidia GTX 1080Ti gpus

| Batch Size | MF | NEUMF | LIGHTGCN | BOX |
|---|---|---|---|---|
| 64 | 08:37 | 17:00 | 70:30 | 19:32 |
| 128 | 04:32 | 09:46 | 38:40 | 11:40 |
| 256 | 02:29 | 04:40 | 20:55 | 05:28 |
| 512 | 01:18 | 02:23 | 10:47 | 02:54 |
| 1024 | 00:40 | 01:20 | 05:24 | 01:12 |

In Table 11, we observe that the MF, being the simplest approach with minimal computational requirements, is consistently the fastest across all batch sizes. At the largest batch size (1024), it achieves the shortest training time of just 00:40. The BOX-based method exhibits training times comparable to NEUMF. However, it is significantly faster than LIGHTGCN, which relies on graph convolutional computations. The iterative message-passing operations required by LIGHTGCN result in considerably higher training times, particularly at smaller batch sizes (e.g., 70:30 at a batch size of 64). As the batch size increases, the training time for BOX embeddings becomes almost as efficient as MF. For instance, at a batch size of 1024, BOX achieves a training time of 01:12, compared to 00:40 for MF. This demonstrates that the computational complexity of box embeddings is of the same order as MF, making it a scalable and efficient choice.

Box embeddings are generally quite fast because the computation of box intersection volumes can be parallelized over dimensions. Note that the training times above use Gumble-Box embeddings, which involve log-sum-exp calculations. However, this could be improved even further at inference time by replacing these soft min and max approximations with hard operators. If such an optimized approach is desired, then training can accommodate this by regularizing temperature. For deployment in industrial set-up, we could take additional steps with Box Embeddings as outlined in (Mei et al., 2022b).

