# OpenReview forum: "A Geometric Approach to Personalized Recommendation with Set-Theoretic Constraints Using Box Embeddings"
_ICML.cc/2025/Conference — ICML 2025 poster_

### Official Review · Reviewer_BTfm · 2025-03-10

**Overall Recommendation:** 3

**Summary:**

This paper proposes a geometric approach to personalized item recommendation using box embeddings. Unlike traditional vector-based methods that struggle with set-theoretic operations, the authors introduce axis-aligned hyperrectangles (boxes) to model users, items, and attributes. This enables logical operations such as AND, OR, and NOT to be represented geometrically. The recommendation task is formulated as set-theoretic matrix completion, where items are recommended based on box containment and intersection.

**Claims And Evidence:**

Yes

**Essential References Not Discussed:**

No

**Experimental Designs Or Analyses:**

1. The authors compare their box embedding model against vector-based baselines like matrix factorization (MF), Neural Matrix Factorization (NeuMF), and LightGCN. The box model shows improved performance on queries with set constraints, highlighting its advantage in those scenarios.

2. Ranking metrics such as Hit Rate@K and NDCG@K are used, which are appropriate for measuring recommendation quality. The box model’s gains on these metrics, particularly for composite queries, support the claim that it better captures user intent for complex filters.

3. An ablation on the use of the Gumbel-softmax (comparing different temperature settings or using a hard version) demonstrates its effect on training stability and performance. This helps justify the inclusion of the smoothing technique.

**Methods And Evaluation Criteria:**

Yes

**Other Comments Or Suggestions:**

1. It would strengthen the paper to explain how a user’s query (in words) would be processed. Are attributes manually selected by the user, or could an NLP model tag the query with the corresponding attributes to form the box query? Providing even a brief discussion or an example would make the approach more concrete.

2. The authors could consider extending their framework to support more complex logical expressions. For instance, how would the model handle a query like (User AND Attribute1) OR (User AND Attribute2), or multiple negations? If the method naturally extends (or if not, what the challenges are), mentioning this would be insightful for readers interested in general logical query support.

3. Including a complexity analysis (even just Big-O) or some timing experiments for query evaluation would be helpful. For example, how does the recommendation runtime grow with the number of items or attributes? Understanding this would indicate the practicality of the approach for large-scale use.

**Other Strengths And Weaknesses:**

Strengths:

1. The conceptual novelty of casting recommendation as a set-theoretic problem and using boxes to represent sets is a significant strength. It offers a new perspective that is both intuitive (via geometry) and potentially powerful for complex preference queries.

2. Empirical validation on multiple datasets strengthens the work. By showing improvements on MovieLens, Last.fm, and NYC-R, the authors demonstrate that the approach is not tailored to a single scenario.

3. The mathematical formulation (energy function and training procedure) is sound and clearly derived. The use of an energy-based model with appropriate smoothing shows careful consideration of the optimization challenges.

Weaknesses:
1. Scalability to more complex queries is not fully explored. While the method works for the queries tested, it’s unclear how performance or accuracy holds up with queries that combine many conditions or involve deeper logical structure.

2. Mapping from natural language to attributes is a practical concern not addressed. In a deployed system, users might issue queries in text (e.g., “find me action movies from the 90s not starring actor X”), which requires translating into the appropriate box intersections. The paper assumes structured attribute inputs, leaving this mapping as future work.

**Questions For Authors:**

1. How does the system handle free-text user queries or preferences? Do you assume a fixed set of attributes that users filter by, or is there a component to dynamically infer attribute constraints from text?

2. Your experiments focus on queries with one or two attribute filters. How would the model scale to queries with multiple attributes and negations (e.g., User AND A1 AND A2 AND NOT A3). Would there be any challenges in training or evaluation with deeper logical compositions?

3. ou chose Gumbel-softmax to approximate the min/max. Did you consider other continuous relaxations (e.g., sigmoid approximations of indicator functions, or other softmin/softmax approaches)? If so, what made Gumbel-softmax the most suitable for this problem?

**Relation To Broader Scientific Literature:**

1. The paper builds on prior work in geometric embeddings for logic and set reasoning. For example, in knowledge graphs, Query2Box embeds complex logical queries as boxes to answer multi-hop queries with AND/OR operators​. Similarly, in NLP semantics, Word2Box shows that box embeddings can capture set relations among word meanings (e.g., "red cars" is a subset of "cars" in embedding space)​. This work extends such ideas to the recommendation domain.

2. It contrasts with traditional collaborative filtering and neural embedding approaches. Standard latent factor models (e.g., matrix factorization or neural CF) represent users and items as point vectors, which struggle to naturally perform set operations. By using regions (boxes) instead of points, this model can directly encode logical constraints that would be hard to achieve with dot-product similarity alone.

**Theoretical Claims:**

The paper provides a clear mathematical framework for representing user preferences and item attributes as boxes. Set operations correspond to geometric operations (intersection for AND, union for OR, complement for NOT), aligning well with set theory. This grounding makes the approach interpretable.

---

> ### Author Rebuttal · Authors · 2025-04-01
>
> We appreciate the reviewer’s suggestion to clarify how natural language queries are mapped to structured forms, as well as their thoughtful concerns regarding generalization, scalability, and efficiency.
>
> **Mapping from natural language to attributes:**
>
>  In real-world recommendation systems — such as streaming platforms, e-commerce sites, or travel services — free-text queries (e.g., “funny action movies without clowns”) are typically mapped to a curated set of item tags (e.g., genre, theme, metadata) via a natural language understanding (NLU) module.
>
> Our model operates **after** this step, assuming a structured query (e.g., `Action ∧ Comedy ∧ ¬Clowns`) has already been derived. While we do not include an NLP module, it is **modular and complementary** to our framework. Recent LLMs (e.g., GPT-4o) can perform this front-end task — identifying attribute constraints from text — while our model serves as the **back-end executor** for set-theoretic reasoning.
>
> We thank the reviewer for this helpful suggestion; including such an example would indeed strengthen the narrative, and we will incorporate a clarifying illustration in the final version.
>
> **Extending the Framework for more complex queries:**
>
> **1. Support for Arbitrary Logical Queries.**
>
> Using the **inclusion-exclusion principle**, any Boolean query can be transformed into disjunctive normal form (DNF). For example, a query like $(\text{User} \wedge A_1) \vee (\text{User} \wedge A_2)$ can be evaluated as:
>
> $\text{Score}(((U \cap A_1) \cup (U \cap A_2)) \cap i) = \text{Score}(U \cap A_1 \cap i) + \text{Score}(U \cap A_2 \cap i) - \text{Score}(U \cap A_1 \cap A_2 \cap i)$, $i$ is the item.
>
> Our **containment-based scoring function** remains applicable to any logical query formed over box embeddings, including negations and disjunctions.
>
> **2. Time complexity / Efficient Implementation via Gumbel Boxes.**
>
> Each term in the DNF expansion involves computing the **intersection of multiple boxes**, which we implement using a **log-sum-exp (LSE)** operation over the min and max coordinates. For a query involving $k$ attributes, the largest term in the DNF requires intersecting $k$ boxes, resulting in a complexity of $\mathcal{O}(k \cdot d)$, where $d$ is the box embedding dimension.
>
> Now to compute scores for the  full Boolean query with $t$ inclusion-exclusion terms: $\mathcal{O}(t \cdot k \cdot d)$. In worst case — e.g., a disjunction over $k$ variables — $t$ can be $\mathcal{O}(2^k)$.
>
> - However, typical queries involve conjunctions and negations, resulting in **far fewer DNF terms** than the worst case. E.g.,(User∧Children∧Comedy∧¬Clowns)∨(Monster∧¬Animation) results into 8 terms in its DNF form.
> - Additionally, **each term is independently computable**, and we parallelize the LSE computation **across dimensions and query terms for** intersection volumes calculation, enabling efficient batched evaluation. These optimizations are provided in our codebase.
>
> There would be no difficulty in operating on more complex queries, such as $U \cap A_1 \cap A_2 \cap \neg A_3$. Note that our model does not need to be **trained on complex queries** — it is trained only on pairwise interactions but naturally generalizes to a broad space of **unseen logical compositions.** As noted above, evaluating on deeper queries is computationally efficient, and so we could certainly evaluate our model on such queries. The main challenge would be identifying deeper queries which would be representative of a real user query distribution, which is non-trivial.
>
> **Why Gumbel max instead of softmax-max:**
>
> We initially explored using the softmax-based approximation for max/min operations, but encountered numerical instability and a lack of probabilistic consistency. Specifically, softmax-max computes a convex weighted average that always underestimates the true max:
>
> $\operatorname{SM-Max}(\mathbf{x}; \tau) =\sum_{i=1}^n\underbrace{
> \Bigl(\frac{e^{x_i/\tau}}{\sum_j e^{x_j/\tau}}\Bigr)}_{w_i}\,x_i.$
>
> $\operatorname{SM-Max}(\mathbf{x}; \tau) < \operatorname{max}(\mathbf{x})$,
>
> $\operatorname{SM-Min}(\mathbf{x}; \tau) = -\operatorname{SMM}(-\mathbf{x}; \tau) > \operatorname{min}(\mathbf{x})$
>
> This becomes problematic when computing box intersections, which rely on min/max operations over interval boundaries. For example, if box $[c,d]$ is fully contained in $[a,b]$ we should have $\operatorname{Vol}([a,b] \cap [c,d]) = \operatorname{Vol}([c,d])$. But the softmax-based approximation inflates the intersection to $[x, y]$ where $x < c$ and $y > d$, violating containment and producing invalid probabilities (e.g., $\operatorname{P}([x,y] \mid [c,d]) > 1$).
>
> In contrast, GumbelBox embeddings have the opposite quality, they upper bound the max operation ensuring the $\operatorname{P}([x,y] \mid [c,d]) < 1$.
>
> We thank the reviewer for highlighting this point — it indeed reflects an interesting theoretical contribution. We will provide the detailed proof in the final version of the paper.

---

### Official Review · Reviewer_Qs1s · 2025-03-14

**Overall Recommendation:** 3

**Summary:**

This paper demonstrates the inconsistency of existing vector embedding models for personalized item recommendation task and discusses the challenges faced by existing machine learning approaches for this problem setup. To alleviate the issue, the authors formulated the problem of personalized item recommendation as matrix completion where rows are set-theoretically dependent. To capture this set-theoretic dependence the authors represented each user and attribute by a hyper-rectangle or box. Box embeddings can intuitively be understood as trainable Venn diagrams, and thus not only inherently represent similarity, but also naturally and faithfully support arbitrary set-theoretic relationships. Queries involving set-theoretic constraints can be efficiently computed directly on the embedding space by performing geometric operations on the representations. An extensive empirical study comparing various vector and box embedding models for the task of set-theoretic query recommendation is conducted and it demonstrates the superiority of box embeddings over vector-based neural methods on both simple and complex item recommendation queries by up to 30% overall.

**Claims And Evidence:**

The claims made in the submission are supported by clear and convincing evidence.

**Essential References Not Discussed:**

No.

**Experimental Designs Or Analyses:**

The experimental design and analysis of this paper are reasonable on the whole, but there are some problems such as query generation bias, single evaluation measurement, lacking transparency of hyperparameter tuning.

**Methods And Evaluation Criteria:**

The methods and evaluation criteria proposed in this paper are reasonable on the whole, but there are the following major problems:
1. This paper mainly uses Hit Rate@k (HR@10, HR@20, HR@50) as an evaluation measurement. Although it can reflect the hit rate of the model in the recommendation list, it cannot measure the recommendation quality comprehensively. Please add more measurements such as NDCG for a comprehensively evaluation.
2. This paper generates personalized query through algorithm 1 and algorithm 2
It is not sufficiently discussed whether the distribution of these queries truly reflects the actual needs of users. For example, some attribute combinations (such as "science fiction" and "documentary") may rarely appear in actual scenes, but the rationality of these combinations is not verified in the paper.

**Other Comments Or Suggestions:**

None.

**Other Strengths And Weaknesses:**

Strengths:
1.An innovative embedding method (box embedding) is introduced to represent users and items as hyperrectangles (i.e. "boxes" in geometric space), breaking through the limitations of traditional vector representations such as matrix factorization.
2.Set theory relations (such as intersection, negation) are directly integrated into the embedded space of the recommendation system, and the modeling problem of complex logical constraints (such as "A and B but not C") is solved. The efficient computation of complex queries is achieved through geometric operations such as box intersection and complement, which is intuitive and theoretically attractive..
3.The authors designed an additional task to learn a state augmentation model that can generate informative embedding to assist the agent in obtaining prior knowledge of the environment and accelerate the convergence of agent networks.
4.The method is designed with mathematical rigor: box embedding is defined by the Cartesian product interval and supports the closed geometry operations of set operations (such as Jaccard similarity calculation), and the theoretical derivation is clear.
5.Experiments have verified the effectiveness of the method, and it can improve over the vector method on simple and complex queries. The experimental design is reasonable and the results are credible.
Weakness:
1.Verify that the experiment includes a personalized recommendation task for a real scenario (e.g., Top-K recommendation), not just a test of a synthetic or specific logical query.
2.The baseline methods employed for comparison in the paper are relatively limited in number and somewhat outdated. This restricts the comprehensiveness of the performance evaluation and may not fully highlight the advantages of the proposed method over more recent advancements in the field.
3.User interests may change over time (e.g., from a preference for "action" to "drama"), but the paper does not indicate how the box can be updated to accommodate such dynamic changes. In real recommendation systems, data usually arrive incrementally (such as real-time user interaction), and the paper does not verify whether the method supports online learning.

**Questions For Authors:**

Please see the above comments.

**Relation To Broader Scientific Literature:**

The key contributions of the paper are closely related to existing scientific literature, especially in the application of box embedding, the processing of set theory queries, and the study of context-aware recommendation systems. The work in this paper not only extends the application range of box embedding, but also fills the gap of the existing recommendation system in handling set theory queries, and provides a new solution for personalized recommendation.

**Theoretical Claims:**

Yes, the theoretical claims are checked.

---

> ### Author Rebuttal · Authors · 2025-04-01
>
> Thank you for the thoughtful and constructive review.
> We appreciate your detailed engagement with our paper and the helpful suggestions on evaluation metrics.
>
> **Recommendation Task / NDCG evaluation / Hyperparameter**
>
> We calculate the NDCG score for the set theoretic queries. Please refer to the tables below.
>
> Movielens-1M dataset
> | Method | $U \cap A$ | $U \cap A_1 \cap A_2$ | $U \cap A_1 \setminus A_2$ |
> | --- | --- | --- | --- |
> | MF-Filter | 13.7 | 12.5 | 13.6 |
> | MF-Product | 15.4 | 16.4 | 15.3 |
> | MF-Geo | 11.0 | 10.3 | 11.2 |
> | NeuMF-Filter | 16.5 | 16.5 | 15.7 |
> | NeuMF - Product | 17.0 | 19.9 | 15.9 |
> | LGCN - Filter | 13.1 | 16.0 | 14.0 |
> | LGCN-Product | 12.0 | 11.8 | 13.4 |
> | Box- Filter | 16.7 | 16.5 | 16.9 |
> | Box - Product | 18.0 | 18.6 | 17.8 |
> | Box - Geometric | **18.6** | **21.5** | **17.9** |
>
> NYCR dataset.
> | Method | $U \cap A$ | $U \cap A_1 \cap A_2$ | $U \cap A_1 \setminus A_2$ |
> | --- | --- | --- | --- |
> | MF-Filter | 9.6 | 10.4 | 10.3 |
> | MF-Product | 12.3 | 15.6 | 14.8 |
> | MF-Geo | 11.5 | 11.9 | 11.7 |
> | NeuMF-Filter | 11.1 | 10.1 | 11.8 |
> | NeuMF - Product | 13.9 | 15.7 | 12.9 |
> | LGCN - Filter | 10.8 | 10.8 | 9.1 |
> | LGCN-Product | 13.4 | 15.8 | 11.1 |
> | Box- Filter | 12.1 | 14.3 | 11.8 |
> | Box - Product | **14.4** | 18.6 | 14.6 |
> | Box - Geometric | **14.4** | **19.1** | **14.9** |
>
> Lastfm dataset also shows similar trends (omitting it for character constraint).
>
> In addition to the logical queries, we also perform a standard recommendation task (i.e. recommend an item for a user), reported in terms of NDCG score (Table 6. in Appendix B2)  on all four datasets where we observe that box embeddings outperform the baselines across all datasets. We posit this is because the box embedding space is more accurately able to capture the item-attribute information.
>
> We provide the hyperparameter search details in Appendix B.2. We use wandb.ai for hyperparameter tuning (See Figure 3. for the hyperparams for Box).
>
> **Baseline consideration:**
>
> We have attempted to compare against formidable baselines which seem most capable of capturing set-theoretic semantics. The baselines chosen—MF, NeuMF, and LightGCN—are standard, well-established methods in recommendation systems. They represent varying levels of neural complexity and address user-item and attribute-item interactions effectively. More recent baselines either do not directly provide any particular advantage for set-theoretic queries, or use significant external information (eg. pretrained LLMs). We feel the selection of baselines covers the existing space fairly well (matrix, neural, and graph-based methods) without diluting the central contribution of introducing a method tailored to set-theoretic compositionally or introducing exogenous information.
>
> **Query Generation Bias:**
>
> We agree that not all attribute combinations reflect realistic user intents, and we explicitly address this concern in our **query generation process** through **statistical heuristics combined with manual inspection**, as described in Section 4.2.2.
>
> Specifically, to ensure **semantic plausibility**, we define **viable and non-trivial attribute pairs** for intersection and difference queries based on two criteria:
>
> 1. **Relevance**: We require that the overlap between the attributes is **greater than random chance**, i.e., $|a_1 \cap a_2| > \varepsilon$, ensuring the pair co-occurs frequently enough to represent realistic co-preference.
> 2. **Non-triviality**: We further enforce that the overlap is **not overly large**, i.e., $|a_1 \cap a_2| < \alpha \cdot \min(|a_1|, |a_2|)$, to avoid trivial combinations where one attribute almost subsumes another (e.g., "fiction" ∧ "sci-fi").
>
> This formulation reflects the **plausibility of attribute combinations**, avoiding unlikely or degenerate cases such as "sci-fi and documentary". In practice, the resulting combinations include interpretable and realistic pairs like ("action", "comedy") or ("children", "not monster"), which align with user behavior patterns. To further ensure the quality of queries, we have manually verified the plausibility of the selected attribute combinations to the best of our knowledge and effort. We also plan to release the full dataset upon paper acceptance to support reproducibility and facilitate future work in this space.
>
> **Changing User preferences:**
>
> Our current work does not explore incremental or online updates to user or item representations. However, it follows the standard recommendation framework, where embeddings are periodically retrained using accumulated user-item and item-attribute interactions — a paradigm widely adopted in practical systems. While we have not implemented streaming updates, such extensions could be supported via fine-tuning on recent data or by leveraging continual learning techniques common in embedding-based models.

---

### Official Review · Reviewer_CkQw · 2025-03-14

**Overall Recommendation:** 3

**Summary:**

1. This paper models the problem of attribute-specific query recommendation as “set-theoretic matrix completion”, where attributes and users are treated as sets of items.
2. It demonstrates the inconsistency of existing vector embedding models for the above task, and establishes box embeddings as a suitable embedding method.
3. It conducts an extensive empirical study comparing various vector and box embedding models for the task of set-theoretic query recommendation, and reports substantial improvements over the vector/neural baselines.

**Claims And Evidence:**

Most claims in the paper are either backed by either explanations or experiments. However, it's better to provide evidence to support claims such as "Even advanced neural network-based approaches, which are designed to capture intricate relationships, have
been shown to struggle with set-theoretic compositionally that underlie many real-world preferences".

**Essential References Not Discussed:**

N/A

**Experimental Designs Or Analyses:**

1. The proposed approach is compared with multiple baselines, including MF, NEUMF and LGCN.
2. Three score aggregation methods are adopted: filter, product and geometric.
3. The experiment results support the superiority of the box-embedding based approach over the baselines.

**Methods And Evaluation Criteria:**

1. The proposed method is based on box embeddings with GumbelMax and GumbelMin approximations.
2. The training objective is based on a noise-contrastive estimation.
3. The benchmark datasets are based on MovieLens 1M and 20M datasets, as well as a subset of the Last-FM dataset. In order to adapt these datasets to the attribute-specific query recommendation problem, the authors utilize various resources and preprocessing steps to construct user and attribute sets, along with personalized simple and complex queries.

**Other Comments Or Suggestions:**

1. Equation 1 last step is wrong.

**Other Strengths And Weaknesses:**

Strength:
1. The paper is well written and easy to understand.
2. It provides a novel framework based on set theory to address attribute-specific query recommendation problem.

Weakness:
1. There are typos in some equations.

**Questions For Authors:**

N/A

**Relation To Broader Scientific Literature:**

The paper may be particularly interesting to communities addressing similar matching problem that can be set based.

**Theoretical Claims:**

The paper doesn't provide any theoretical claims or proofs.

---

> ### Author Rebuttal · Authors · 2025-04-01
>
> Thank you for going through the paper so meticulously — we truly appreciate the careful reading and thoughtful feedback. We're glad you found the framework novel and the paper well-written, and we appreciate your comments highlighting both the strengths and areas for improvement.
>
> We apologize for the oversight in the last step of the equation — we’ve corrected it, and the fix will be reflected in the final draft.
>
> **Regarding supporting evidence for the limitations of vector-based embeddings in handling set operations:**
>
> We found QUEST (Malaviya et al., 2023) to be a closely related study that introduces a benchmark for entity-seeking queries with implicit set-based semantics. Although QUEST does not focus on explicit constraints or personalization, which are central to our work, it empirically demonstrates that vector-based T5 embeddings struggle with set-theoretic operations — particularly with negation queries, and often treat conjunctions as disjunctions.

---

### Official Review · Reviewer_onQN · 2025-03-15

**Overall Recommendation:** 2

**Summary:**

This paper proposes a box embedding-based recommendation model that captures set-theoretic constraints in personalized recommendation tasks. Traditional vector-based embeddings struggle with expressing complex relationships like intersection and negation (e.g., "comedy and action but not romance"). The paper introduces an approach using box embeddings—n-dimensional hyperrectangles—to model user preferences and item attributes in a way that naturally supports set-theoretic operations.

The key contributions include:

Formulating recommendation as set-theoretic matrix completion, where user-item interactions are represented via geometric set operations rather than linear factorization.
Box embedding representations that allow efficient and faithful modeling of intersection, negation, and containment relationships.
A novel training method that optimizes embeddings using a probabilistic volume function, ensuring semantic consistency with set-theoretic operations.
Extensive empirical evaluation on MovieLens, Last.fm, and NYC restaurant datasets, showing that box embeddings outperform matrix factorization (MF), neural matrix factorization (NeuMF), and graph-based methods (LightGCN) by up to 30% in set-theoretic query performance.

**Claims And Evidence:**

Claim 1: Box embeddings better model set-theoretic constraints compared to vector-based methods.
Supported by evidence: Empirical results in Table 3 show that Box-Geometric consistently outperforms vector-based models (MF, NeuMF, LightGCN) in Hit Rate@10/20/50.

Claim 2: Box embeddings enable efficient computation of intersection and negation operations.
Supported by evidence: The paper introduces a probabilistic volume-based approach (Equation 2) that ensures intersection and containment scores are computed efficiently.

Claim 3: The model generalizes well to unseen user-item interactions.

Partially supported: The Generalization Spectrum analysis (Table 4) shows that box embeddings exhibit a smaller performance gap when transitioning from weak to strong generalization settings.

Potentially problematic claim:
The authors suggest that traditional vector embeddings inherently lack set-theoretic compositionality, but some recent models (e.g., graph-based hybrid models) can partially capture these relationships—a comparison with these approaches is missing.

**Essential References Not Discussed:**

1. The paper does not cite recent work on hybrid graph embeddings for reasoning over structured constraints.
2. LLM-based recommendation systems could be discussed as an alternative for compositional queries.

**Experimental Designs Or Analyses:**

Experiments systematically evaluate:
1. Simple queries (e.g., “Bob wants a comedy movie”).
2. Complex queries (e.g., “Bob wants a comedy & action movie but NOT romance”).

Generalization spectrum study:
1. Tests how well the model handles unseen user-item interactions.
2. Box embeddings retain higher accuracy than MF/NeuMF when tested on unseen combinations.

Baselines include:
1. Traditional MF, NeuMF, LightGCN.
2. Three aggregation methods: filtering-based, product-based, and geometric.

Potential improvement:
Since the industry uses the user history sequence to build the recommendation system. Is this MF method still a solid baseline?

**Methods And Evaluation Criteria:**

Box embedding formulation:
1. Represents users, items, and attributes as hyperrectangles in a latent space.
2. Computes recommendation scores using geometric containment relationships.

Training objective:
1. Optimizes embeddings with a noise-contrastive estimation loss, ensuring learned representations respect set-theoretic operations.
Evaluation metrics:
2. Hit Rate (HR@k) and NDCG@k for ranking-based evaluation.

Benchmarking against:
1. Matrix Factorization (MF), Neural Collaborative Filtering (NeuMF), and Graph Convolutional Networks (LightGCN).
2. Experiments on three datasets (MovieLens, Last.fm, NYC-R) with varying sparsity levels.

Potential improvement:
If the recommendation query is implicit, that is, the platform is only equipped with the user interactive history, how this method been applied rather than the current given query pattern.  Since the implicit recommendation is more common.

**Other Comments Or Suggestions:**

1. Would combining box embeddings with GNNs or LLMs improve performance?

2. Investigate real-time feasibility: Can this be efficiently deployed in interactive recommendation scenarios?

3. Consider user personalization beyond set-theoretic constraints: How can temporal user behavior be incorporated?

**Other Strengths And Weaknesses:**

Strengths

1. Conceptually novel: Reformulates recommendation as set-theoretic matrix completion.
2. Box embeddings naturally handle complex queries.
3. Empirical validation: Large-scale experiments show clear performance gains.
4. Probabilistic smoothing improves training stability.

Weaknesses

1. No real-world user study: Effectiveness is validated only through automated ranking metrics.
2. Limited generalization beyond movie/music domains: Would it work for news, e-commerce, or job recommendations?
3. Cold-start problem remains: The approach still requires enough observed user-attribute interactions. How to solve the cold Start problem in this set embedding method

**Questions For Authors:**

1. Box intersection operations could become expensive in high-dimensional spaces.
2. How does it compare to hybrid graph-based approaches?
3. Could it handle temporal or evolving user preferences?
4. Can the box embeddings adapt over time to user behavior shifts?
5. Would this approach work for content-aware recommendations?
6. Could text/image embeddings be incorporated into the box representation?

**Relation To Broader Scientific Literature:**

Builds on prior work in box embeddings:
1. Extends prior probabilistic box models used in knowledge graph reasoning to recommender systems.
2. Connects to compositional reasoning models:

Challenges traditional word2vec-style additive composition by demonstrating superior set-theoretic consistency.

**Theoretical Claims:**

Seems solid.
1. Set-theoretic matrix completion formulation ensures the inductive bias aligns with query constraints.
2. Probabilistic volume estimation replaces hard min/max operations with smooth approximations to enable gradient-based training.
Containment-based scoring function is proven to respect set relationships in Equations (1)–(3).

---

> ### Author Rebuttal · Authors · 2025-04-01
>
> We thank the reviewers for raising thoughtful and forward-looking questions regarding possible extensions of our approach.
>
> We agree that a recommendation systems framework should be recognized not only for its current performance but also for its extensibility to address practical challenges such as deployability, handling cold start problem, content-aware recommendation, and evolving user behavior etc. While we focus our paper on a clearly defined core contribution of handling set-theoretic constraints, we touch upon these broader aspects in the rebuttal to highlight the extensibility and practical relevance of our framework. We will include these discussions in the final version of the paper.
>
> **Box operation in high dimensions and Deployment in a search system:**
>
> Box embeddings remain computationally efficient even in high dimensions, as box intersection volume calculation can be parallelized across dimensions. Our method uses GumbelBox embeddings during training, where soft intersections are computed via log-sum-exp operations, however inference speed can be further improved by substituting soft approximations with hard min/max operations, providing faster evaluation at deployment time.  This is achievable by annealing the Gumbel temperature to promote sharper boundaries during training (Capacity & Bias). Due to these advantages, box embeddings can actually provide even faster inference than approximate k-NN algorithms on vectors, as demonstrated in Mei et al. (2022b) which significant improvement over FAISS.
>
> **Content-aware recommendation / Handling Text image as side information / Cold start problem.**
>
> In this work, we treat genres, tags, and metadata as item content, making our setup inherently content-aware with respect to structured attributes. We understand the reviewer’s interest in whether our box embedding framework can be extended to richer content modalities such as text or images.
>
> A practical approach is to encode text/image features using pretrained models like BERT or CLIP, and then project them into the box embedding space via a learnable transformation. Prior work supports this, to give a few examples; Onoe et al. (2021)  derived box embeddings for entity mention representation from BERT,  Rau et al. (2020) use ResNet50 features to construct box embeddings for visual scenes. Such multimodal integration would enable our model to handle unstructured content and is particularly promising for cold-start scenarios.
>
> While our current method assumes some user/item interactions during training (as in standard collaborative filtering), we are actually uniquely well-suited to handle cold-start settings. To do so, we could “pretrain” item representations such that they are contained within their attribute boxes. In typical cold-start fashion, a new user would generally provide a list of like/dislike annotations on attributes, which would directly correspond to set-theoretic operations on the attribute box representations.
>
> **Evolving user preferences over time:**
>
> Our current work does not explore incremental or online updates to user or item representations. However, our method can operate within the standard recommendation framework, where embeddings are typically retrained periodically using updated user-item and item-attribute interaction data. This batch retraining paradigm is widely adopted in practical systems, where new interactions are accumulated over time and incorporated during scheduled model updates. While we have not implemented online or streaming updates in this work, such extensions could be supported by fine-tuning user or item embeddings on recent interaction data.
>
> **Recommendation without constraint (implicit setting):**
>
> In the implicit setting, the system defaults to using the user's historical preferences — captured via the user box learned from past interactions — as the query itself. This allows our model to function like standard collaborative filtering approaches, recommending items based on geometric containment between the user box and item points in the latent space.
>
> **Comparison to hybrid graph-based approaches:**
>
> We acknowledge that many recent *attribute-aware* recommendation models use Graph Convolutional Networks (GCNs) to capture item-user-attribute interactions. In our work, we include **LightGCN** (He et al., 2020) as a baseline and adapt it to a *hybrid graph* where items are connected to both users and attribute tags as a form of graph. This ensures attribute information is incorporated, aligning with our study’s objectives. We agree that such hybrid graph-based methods are strong baselines and may offer partial inductive biases for compositional reasoning. Note, however, that our method outperforms LightGCN by >40% on average over all the query types.
>
> If the reviewer has a specific hybrid graph-based method in mind that supports logical constraints or compositional queries, we would be happy to review and discuss it in the final version.

---

### Decision · Program_Chairs · 2025-05-01

**Decision:**

Accept (poster)

**Comment:**

This paper presents a geometric approach to recommender systems using box embeddings where users, items, and attributes are all represented as boxes in the latent space to facilitate set-theoretic operations, such as negation and intersection, e.g., recommending a comedy but not romance.

Overall the paper received positive-leaning scores where all reviewers praise the innovative usage of the box embeddings in the context of recommender systems. There are some shared concerns on the baseline selections. Overall I believe the positive overweights the negative. I also skimmed the paper myself (albeit likely not as carefully as a reviewer would). I have one comment: the authors claim that the item-attribute matrix tends to be incomplete (just like user-item matrix), hence an imputation approach should be used here to create an "imputed" attribute for filtering. However, I think in practice, the item-attribute matrix can often be considered "complete" since they are often curated manually. Furthermore, I don't think the underlying assumption behind matrix completion for user-item matrix (user A and B interacted with similar items, hence recommending items that user A interacted with but not B to B is a good idea) necessarily translates to item-attribute matrix -- if two movies A, B have similar attributes, it doesn't mean B should also have the additional attributes that only applies to A. Because of this, it seems to me that the most natural baseline would be simply do hard filtering.

Another thing that I want to point out is that due to the unique topic combinations involved in this paper, none of the reviewers are actually very familiar with the box embedding related literature (myself included). Therefore our assessment might be less confident than normal. I am still voting for accept but open to be bumped down.